# Manipulation as in Simulation: Enabling Accurate Geometry Perception in Robots

**Minghuan Liu**[1,†,*]**, Zhengbang Zhu**[1,2,*]**, Xiaoshen Han**[1,2,*]**, Peng Hu**[1,*]**, Haotong Lin**[1,3]**, Xinyao Li**[2]**,
Jingxiao Chen**[1,2]**, Jiafeng Xu**[1]**, Yichu Yang**[1]**, Yunfeng Lin**[2]**, Xinghang Li**[4]**, Yong Yu**[2]**,
Weinan Zhang**[2,†]**, Tao Kong**[1]**, Bingyi Kang**[1,†]

[1]ByteDance Seed, [2]Shanghai Jiao Tong University,
[3]Zhejiang University, [4]Tsinghua University
ericliuof97@gmail.com, wnzhang@sjtu.edu.cn, bingykang@gmail.com

## Abstract

Modern robotic manipulation primarily relies on visual observations in a 2D color space for skill learning but suffers from poor generalization. In contrast, humans, living in a 3D world, depend more on physical properties-such as distance, size, and shape-than on texture when interacting with objects. Since such 3D geometric information can be acquired from widely available depth cameras, it appears feasible to endow robots with similar perceptual capabilities. Our pilot study found that using depth cameras for manipulation is challenging, primarily due to their limited accuracy and susceptibility to various types of noise. In this work, we propose Camera Depth Models (CDMs) as a simple plugin on daily-use depth cameras, which take RGB images and raw depth signals as input and output denoised, accurate metric depth. To achieve this, we develop a neural data engine that generates high-quality paired data from simulation by modeling a depth camera's noise pattern. Our results show that CDMs achieve nearly simulation-level accuracy in depth prediction, effectively bridging the sim-to-real gap for manipulation tasks. Notably, our experiments demonstrate, for the first time, that a policy trained on raw simulated depth, without the need for adding noise or real-world fine-tuning, generalizes seamlessly to real-world robots on two challenging long-horizon tasks involving articulated, reflective, and slender objects, with little to no performance degradation. We hope our findings will inspire future research in utilizing simulation data and 3D information in general robot policies. We release the dataset, models for various depth cameras, along with an easy-to-use guide for sim-to-real transfer at https://manipulation-as-in-simulation.github.io/.

## 1 Introduction

Manipulation is a fundamental capability expected of robots, primarily involving skilled interactions with diverse objects, and thus necessitating visual observations. Recent advances show that robots can perform various tasks using 2D color images from single or multiple viewpoints (Chi et al., 2023; Zhao et al., 2023; Fu et al., 2024; Li et al., 2024a; Wu et al., 2024; Team et al., 2024; Kim et al., 2025; Black et al., 2024). While color images provide rich semantic information, humans operate in a 3D world and rely on geometric cues—such as shape and spatial relationships—to distinguish objects (e.g., bottles versus bowls) and comprehend the skills required. This reliance on geometry, rather than texture, enables functional inference and precise interaction with objects.

With the widespread availability of depth cameras, acquiring 3D geometric information appears to be straightforward, suggesting that robots could be endowed with similar perceptual capabilities (Fang et al., 2023; Yan et al., 2024; Liu et al., 2024; Ze et al., 2024b). However, their unreliable output, frequent mode failures, and sensitivity to noise pose significant challenges. Although recent studies have integrated 3D representations into robotic manipulation, performance remains limited by the poor quality of depth data produced by such devices. Consequently, evaluations are typically

---

*Equal contribution. †Corresponding authors.

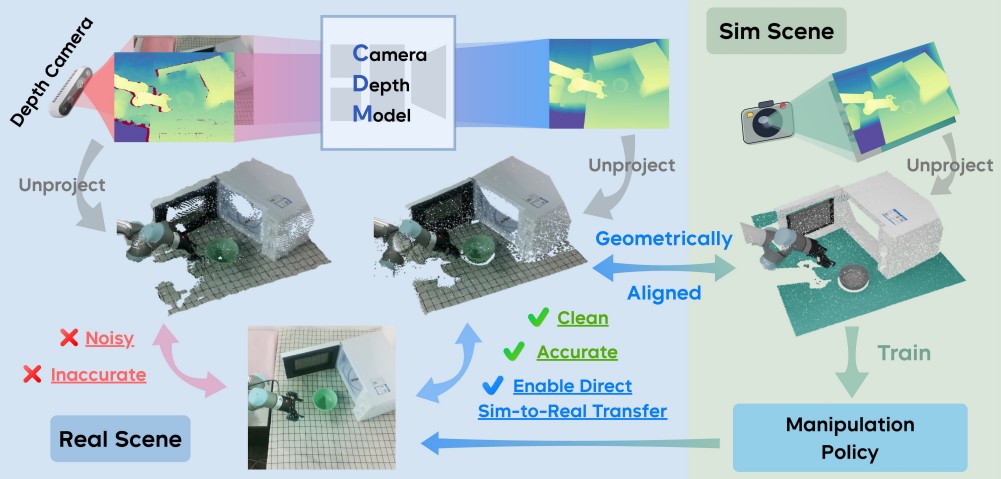

Figure 1: **How the proposed camera depth model (CDMs) makes real-world manipulation as in simulation.** The illustration is made on the RealSense D435 camera with CDM-D435. With CDM, the manipulation policy learns from accurate geometric information, which is aligned between the simulation and the real world.

restricted to simulation environments (Zhen et al., 2024; Zhu et al., 2024), where clean and accurate depth is available; or rely on downsampled point clouds (Ze et al., 2024a; Hua et al., 2024; Ze et al., 2024b) to mitigate noise in real-world scenarios. As illustrated in Fig. 1, real-world depth camera data often contains significant and characteristic noise artifacts, resulting in inaccurate perception of objects and environments by robots.

To mitigate the fundamental problem of depth perception and bring accurate geometry into robotic manipulation, this paper proposes camera depth models (CDMs), a plug-in solution for depth cameras that enhances geometric accuracy, as illustrated in Fig. 1. A CDM processes RGB images and noisy depth signals from a specific depth camera to produce high-quality, denoised metric depth. To train such models, we developed a multi-camera mount and collected a dataset of RGB-depth pairs from seven cameras across ten depth modes. Leveraging both this dataset and open-source simulated data, we designed a neural data engine that models the noise patterns of depth cameras to generate high-quality paired data in simulation. To address the scale mismatch in synthesized noise, we propose a novel guided filter approach for noise augmentation. CDMs achieve nearly simulation-level 3D accuracy, effectively bridging the sim-to-real geometry gap from the real-world perspective.

Our experiments evaluate CDMs in real-world imitation and sim-to-real manipulation tasks. We show that CDMs enable robot policies to learn generalizable skills from accurate geometric information. Notably, we demonstrate, for the first time, that a policy trained on raw simulated depth, without the need for adding noise or real-world fine-tuning, can transfer seamlessly to real robots on two challenging long-horizon tasks involving articulated, reflective, and slender objects. These results highlight the potential of CDMs to leverage simulation and underscore the importance of accurate geometric data in developing robust, generalizable robot policies.

In a nutshell, the contributions of this paper are:

1. We introduce ByteCameraDepth, a real-world multi-camera depth dataset comprising over 170,000 RGB-depth pairs captured by seven depth cameras.
2. We proposed and released a family of camera depth models (CDMs), a plug-in solution that enhances depth perception accuracy for widely used depth cameras.
3. Through CDMs, we demonstrate how the sim-to-real geometry gap can be bridged, highlighting the critical role of accurate geometric information in robotic manipulation tasks.

## 2 RELATED WORK

**Metric depth prediction.** Recent advances in depth-fundamental models, such as the Depth Anything (DA) series (Yang et al., 2024a;b), has significantly advanced scene geometry estimation and

high-resolution relative depth prediction across diverse open-world images, showing strong generalization. However, most real-world applications require metric depth rather than relative depth. Simply fine-tuning DA models to predict metric depth (Yang et al., 2024b) remains constrained by a fixed depth scale and is susceptible to scale ambiguities (Yin et al., 2023; Hu et al., 2024). Although recent approaches (Wang et al., 2024b; 2025b) introduce affine-invariant techniques to train relative models on large-scale datasets and achieve improved metric depth estimation via post-processing with a prompt depth, the fundamental scale ambiguity in monocular images, especially for scenes with large depth ranges, remains unresolved. To address this, many recent approaches choose to incorporate explicit scale cues for predicting metric depths. For instance, Guizilini et al. (2023) and Piccinelli et al. (2024) introduce camera intrinsics into the model. Lin et al. (2025) and Wang et al. (2025c) proposed a more straightforward way that directly integrates scale information by prompting paradigms, *i.e.*, low-quality depth images or sparse LiDAR signals, into the model's architecture of the pre-trained DA model, and finetuned the model on RGBD datasets with handcrafted prompt depth images on synthesized data. Nevertheless, they are limited in prompt images made with the style of handcrafted rules and are hard to work well on diverse sensor configurations for dynamic scenes. In addition to these solutions with a depth prompt, there are some works focused on recovering metric depth (disparity) from stereo RGB images (Wen et al., 2025), which provide implicit depth cues through disparity, but they often require careful calibration and are limited in diversity. Such methods are limited to working on stereo cameras (and with RGB only) and require the precise camera intrinsics (baseline distance, focal length, and so on) to obtain the depth.

**Manipulation with 3D representation.** Robotic manipulation requires accurate perception of object states. Classical planning-based approaches typically depend on a calibrated perception module to identify the 3D positions of relevant objects, which are then used to plan feasible manipulation paths (Fang et al., 2023). Learning-based methods, on the contrary, focus on modeling autonomous robot policies using neural networks. Most recent works rely on RGB images, ranging from single-view (Chi et al., 2023; 2024; Kim et al., 2025) to multi-view setups (Zhao et al., 2023; Fu et al., 2024), and from single-task policies (Chi et al., 2023; Zhao et al., 2023) to multi-task generalist policies (Li et al., 2024a; Wu et al., 2024; Kim et al., 2025; Black et al., 2024). While these methods achieve strong results, they struggle to generalize across visual conditions. To better leverage scene geometry, recent works incorporate 3D representations into policies in different ways: some use point clouds to improve generalization across objects with similar shapes but varying textures and backgrounds (Ze et al., 2024a;b; Hua et al., 2024), though these approaches typically require careful calibration, downsampling, and cropping to mitigate sensor noise; some train depth-only policies that segment objects and mitigate sim-to-real discrepancies (Liu et al., 2024); and others propose 3D-aware foundation models that jointly process depth, point clouds, or bounding boxes, though these are often limited to simulation with perfect 3D perception (Zhen et al., 2024; Zhu et al., 2024). Furthermore, some works explore more complex representations, such as neural radiance fields (Li et al., 2022; Yan et al., 2024) and dense voxels (Shridhar et al., 2023; Ze et al., 2023); however, all of these 3D representations ultimately depend on transforming the original camera depth into real-world scenarios.

## 3 Camera Depth Models

Existing depth foundation models can estimate proper relative depth without a geometric prior. However, for real-world manipulation tasks, models must predict absolute metric depth. This requires two key capabilities: 1) identifying semantically meaningful local regions for objects and backgrounds in RGB images, and 2) assigning accurate metric depths to these regions using coarse depth prompts from camera depth images. Notably, this task extends beyond simple denoising or depth completion, as raw depth readings from various depth cameras exhibit diverse working ranges, failure modes, noise patterns, and biases. Fig. 4 provides a glimpse of the noisy depth images produced by consumer-grade depth cameras.

### 3.1 Noise from Depth Cameras

We categorize two general types of noise, *i.e.*, the value noise and the hole noise, as depicted in Fig. 2-left. Intuitively, hole noise manifests as missing data in depth readings, often caused by depth estimation algorithms (e.g., stereo matching) or environmental factors, such as lighting or material

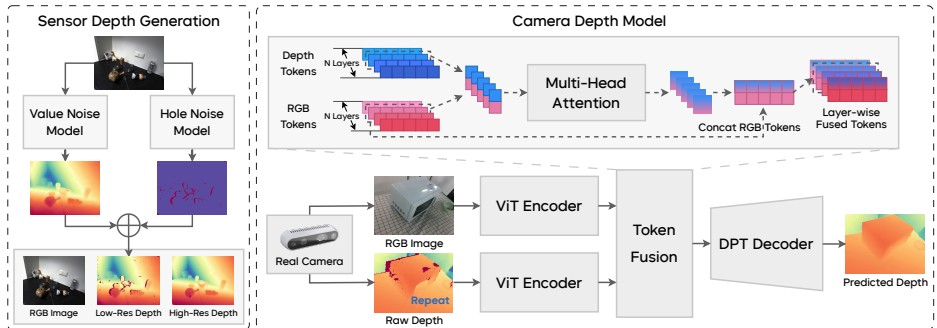

Figure 2: **Overview of camera depth models.** Left: camera depth generation to synthesize the datasets for training camera depth models, where the value/hole noise models are trained using the collected dataset. (Sec. 3.4) Right: the camera depth model, which is built on two ViT encoders (Dosovitskiy et al., 2020) and fine-tuned from a depth foundation model (Yang et al., 2024b); the RGB and depth tokens are fused before being given to a DPT decoder (Ranftl et al., 2021). Such a structure allows the model to receive sparse depth from sensors for prediction without any pre-processing like hole-filling. (Sec. 3.2)

properties. Value noise encompasses all other inaccuracies, including biases specific to each camera, as well as blur, jitter, and other distortions. For instance, stereo matching-based cameras often produce holes around object boundaries, while LiDAR-based cameras struggle with black or highly reflective surfaces. Both types perform poorly on transparent or mirror-like objects, such as glass. These noise patterns depend on the camera's intrinsic parameters and physical installation.

Therefore, develop an effective metric depth model for a specific camera, the model must 1) refine coarse, low-quality depth prompts from the camera into precise metric depth estimates, while 2) correcting faulty depth readings by leveraging semantic information from RGB images. Balancing reliance on sensor data with skepticism of its inaccuracies poses a significant challenge, making a generalizable solution nontrivial. This motivates the development of camera-specific depth models (CDMs) tailored to individual depth cameras.

## 3.2 MODEL DESIGN

We designed our CDMs $M$ for specific depth cameras to take a pair of an RGB image $\mathbf{I} \in \mathbb{R}^{3 \times H \times W}$ and a depth image $\mathbf{D} \in \mathbb{R}^{H \times W}$ from the depth camera, and predict a high-quality metric (absolute) depth image $\hat{\mathbf{D}} \in \mathbb{R}^{H \times W}$. The proposed model structure of CDMs is illustrated in Fig. 2-right. In particular, we design a dual-branch ViT (Dosovitskiy et al., 2020) architecture to achieve the above-mentioned capabilities by separately capturing semantic information from the RGB and the depth images, as well as scale information that is cross-modal but aligned in feature tokens $X$:

$$X^{\mathbf{I}} = \text{ViT}^{\mathbf{I}}(\mathbf{I}), X^{\mathbf{D}} = \text{ViT}^{\mathbf{D}}(\mathbf{D}) , \tag{1}$$

where $X^{\mathbf{I}} = \{X_1^{\mathbf{I}}, \cdots, X_N^{\mathbf{I}}\}$ and $X^{\mathbf{D}} = \{X_1^{\mathbf{D}}, \cdots, X_N^{\mathbf{D}}\}$ are feature tokens encoded by the RGB branch and the depth branch, separately.

Subsequently, we fuse these two types of information through a feature token fusion module. Since the token fusion module primarily serves to augment semantic information with scale information, it is only necessary to fuse tokens corresponding to the same spatial locations. Based on this, the fusion module only performs self-attention on corresponding tokens to accomplish bidirectional feature fusion, and results in depth features $\tilde{X}$ imbued with scale information:

$$\tilde{X} = \sum_i \text{MHA}(\{[X_i^{\mathbf{I}}; X_i^{\mathbf{D}}]\}_{i=1}^N) , \tag{2}$$

where MHA stands for multi-head attention, and $[;]$ is the concatenation operation. The entire fusion process occurs across multiple levels of feature tokens, allowing for deeper integration and the ability to incorporate both global- and local-scale information, especially when the camera depth prompt has large missing regions, where global-scale information is particularly needed.

Additionally, we concatenate the original RGB feature tokens into the fused feature tokens. These fused feature tokens, along with the RGB feature tokens, are concatenated to prevent loss of semantic

information, and then passed through a DPT head (Ranftl et al., 2021) to produce scale-aware depth estimation results $D$.

$$\hat{\mathbf{D}} = \text{DPT}([X^{\mathbf{I}}; \tilde{X}]) \,. \tag{3}$$

Compared with previous works that fuse the prompted depth information simply in the shallow decoding phase (Wang et al., 2025c; Lin et al., 2025), our proposed CDM structure provides a much more informative representation of the depth feature and its alignment with the RGB feature, and thus can perceive the raw depth image directly, without preprocessing such as hole-filling Wang et al. (2025c); Lin et al. (2025). Through the simple but representative structure design, CDM simplifies the inference procedure, provides the metric depth, and works as a simple plugin after the camera input, thereby fulfilling the three desiderata mentioned in the beginning.

### 3.3 BYTECAMERADEPTH: A MULTI-CAMERA DEPTH DATASET

To train our CDMs, we will need a dataset that contains triplets, *i.e.*, RGB image $I$, low-quality depth image $D$, and ground-truth depth image $\overline{\mathbf{D}}$. However, the low-quality depth images, although they have usually been handcrafted by adding typical noise patterns to ground-truth depth images that are simulated by basic depth measurement principles, other factors are hard to be accurately modeled in the simulation, for instance, the camera parameters, the implementation and optimization details in each depth camera hardware and software are case by case, resulting distinct noise behaviors. On the contrary, we can easily collect low-quality depth data from real sensors, but it is hard to get perfect depth data. Therefore, naturally, we propose to learn the noise pattern with neural networks automatically from real-world data, and then synthesize the noisy low-quality depth image with the learned noise models.

To this end, we collect typical depth patterns and construct a dataset for various depth cameras that are commonly used in daily robot experiments. Specifically, our dataset spans 10 depth modes from 7 different depth cameras, including different stereo and lidar cameras. To achieve highly efficient data collection, we design a multi-camera mount device to capture data simultaneously, details shown in Appendix C. Our datasets contain more than 17,000 images for each camera, sampled from videos at 5Hz, covering 7 different scenes, including kitchens, living rooms, markets, bedrooms, bathrooms, offices, and breakrooms. A visual glimpse of the dataset is shown in Appendix C.

### 3.4 DATA SYNTHESIS WITH NOISE MODELS

We train two noise models on our collected depth dataset for each camera, which are then used for generating stylized low-quality depth images on open datasets to train CDMs.

**Hole noise model.** We treat the hole noise prediction as a binary-class prediction given the RGB image $I$, thereby training the hole noise model $N_{\text{hole}}$ with a pretrained DINOv2 backbone (Oquab et al., 2023) with a DPT head (Ranftl et al., 2021) to predict the valid mask (*i.e.*, hole/non-hole) for each pixel on the camera depth $\mathbf{D}$. Formally, this corresponds to optimizing the objective:

$$\ell(N_{\text{hole}}(\mathbf{I})) = \sum_{i=0,j=0}^{i=H,j=W} [y_{i,j} \log \sigma(x_{i,j}) + (1 - y_{i,j}) \log(1 - \sigma(x_{i,j}))] \,, \tag{4}$$

where $x_{i,j} = N_{\text{hole}}(\mathbf{I})_{i,j}$ denotes the $i,j$-th pixel on the mask image predicted by the hole noise model $N_{\text{hole}}$, $y_{i,j} = \mathbb{I}(\mathbf{D}_{i,j} = 0)$ denotes if the $i,j$-th pixel corresponds to a hole, and $\sigma$ is the sigmoid function.

**Value noise model.** Motivated by the fact that depth foundation models are predicting a clean style depth image, we regard the value noise prediction as a stylized relative depth prediction problem, thereby turning to the help of Depth Anything V2 (Yang et al., 2024b) (DAV2) by taking the low-quality depth image as the labels for prediction. The training objective of the value noise model $N_{\text{value}}$ is an $L_1$ loss of the predicted depth $\hat{\mathbf{D}}_{\text{value}} = N_{\text{value}}(\mathbf{I})$ and a normalized ground truth depth $\hat{\mathbf{D}}$:

$$\ell(N_{\text{value}}) = L_1(f(\overline{\mathbf{D}}), \hat{\mathbf{D}}_{\text{value}}) \,, \tag{5}$$

where $f$ is the normalization function, in our project, we use the affine-invariant transformation proposed in Wang et al. (2025a). To make sure the value noise can learn proper relative scales

during the data synthesis stage, we also fine-tune the DAV2 model on the synthesized dataset before turning it into a value noise model.

**Synthesizing camera depth.** Having the noise models, we can synthesize the noisy low-quality data on open synthesized datasets with clean ground truth depth. Denote the hole noise model as $H$ and the value noise model as $V$, we cured the synthesized noisy data $\tilde{D}$ given the RGB image $I$ via:

$$\tilde{\mathbf{D}} = \mu(V(\mathbf{I})) * (H(\mathbf{I}) < 0.5) , \tag{6}$$

where $\mu$ is the affine-invariant unscaling function (Wang et al., 2025a) recovering the metric of the predicted relative value noise, referring to the metric of the ground truth depth.

## 3.5 CDM Training

Although we can synthesize training data for specific cameras, and these two types of noise models can learn similar noise patterns compared to the raw depth obtained from the real depth camera, we observed several problems, especially with the value noise models.

**Guided filter for value noise.** A key challenge of the value noise model is its struggle to maintain the correct metric scale on synthesized datasets after fine-tuning on the ByteCameraDepth dataset. This leads to metric discrepancies between the synthesized camera depth and ground-truth depth, causing the trained CDMs to underutilize the metric information in the camera depth prompt. To address this, we propose using the guided filter (He et al., 2012), which assumes the output image $B$ is a local linear transformation of the guided image $G$: $b_i = x_k g_i + y_k$ , where $b_i, g_i$ are the $i$-th pixel of $B$ and $G$, respectively; $x_k, y_k$ are the scale and shift parameters in the kernel window. These parameters are optimized by minimizing the error of the transformation between the input image $A$ and the output image $B$:

$$\sum_{i \in \omega_i} \left( (x_k g_i + y_k - a_i)^2 + \epsilon x_k^2 \right) , \tag{7}$$

where $\epsilon$ is a regularization term. In our approach, the guided filter uses the value noise as the guidance image $G$ and the ground-truth depth as the input image to be filtered $A$. The resulting image $B$ preserves the geometry and structure of the value noise while approximating the correct metric scale across the image. As the kernel size $k$ increases, the output image retains more of the noise structure and less of the ground-truth metric information. To balance this, we employ a randomized kernel size $k$ (ranging from small to large) as an augmentation strategy for the value noise before adding hole noise, which yields optimal results. Additionally, adjusting the maximum value of $k$ controls the model's reliance on the prompt depth: a smaller $k$ aligns the prompt closer to the ground truth, encouraging the model to depend more on the prompt. Furthermore, both noise models struggle to capture high-frequency noise patterns due to the neural network's limitations and our DAV2-like training strategy. To address this, we introduce high-frequency noise via handcrafted rules as an additional augmentation strategy.

**Training loss.** Referring to Lin et al. (2025), we use the $L_1$ loss combined with the gradient loss for better edge depth to train our CDMs $M$, given image $\mathbf{I}$ and its raw depth $\mathbf{D}$:

$$\ell(M) = L_1(\overline{\mathbf{D}}, \hat{\mathbf{D}}) + \ell_{\text{grad}}(\overline{\mathbf{D}}, \hat{\mathbf{D}}) , \tag{8}$$

$$\text{where } \ell_{\text{grad}}(\overline{\mathbf{D}}, \hat{\mathbf{D}}) = (|\tfrac{\partial(\hat{\mathbf{D}} - \overline{\mathbf{D}})}{\partial x}| + |\tfrac{\partial(\hat{\mathbf{D}} - \overline{\mathbf{D}})}{\partial y}|) . \tag{9}$$

During our training, we use disparity as the training target. The weights of the ViT encoder in the RGB and the depth branch are both initialized from DINO-v2 (Oquab et al., 2023), and the decoder is trained from scratch. For the single-channel depth images, by default, they are copied three times before being fed into the network. We synthesized our training data on four simulated datasets: HyperSim (Roberts et al., 2021), DREDS (Dai et al., 2022), HISS (Wei et al., 2024), and IRS (Wang et al., 2019), in a total of 280,000+ images.

## 3.6 Sim-to-Real Manipulation through CDMs

Our camera depth models (CDMs) allow us to obtain a 'simulation-like' depth image in the real world, which provides accurate geometry information. Therefore, simulation data can be fully utilized to learn a manipulation policy, which can be seamlessly transferred to the real world. To

| Split | Filled / Holed / | L1 ↓ | RMSE ↓ | AbsRel ↓ | $\delta_{0.5}$ ↑ | $\delta_1$ ↑ |
|---|---|---|---|---|---|---|
| D435
(IR Stereo) | **Ours (CDM-D435)** | **0.0258** | **0.0404** | **0.0312** | **0.9842** | **0.9951** |
| | **Ours (CDM-L515)** | **0.0182** | **0.0338** | **0.0217** | **0.9877** | **0.9956** |
| | PromptDA*(435) | 0.0434 | 0.0666 | 0.0599 | 0.9459 | 0.9770 |
| | PromptDA*(515) | 0.1830 | 0.2387 | 0.2750 | 0.8802 | 0.9186 |
| | PromptDA | 0.1703 | 0.2971 | 0.2437 | 0.6704 | 0.7229 |
| | PriorDA | 1.2031 | 0.6856 | 1.2030 | 0.0837 | 0.1717 |
| | PromptDA | 0.0396 | 0.0691 | 0.0484 | 0.9503 | 0.9772 |
| | PriorDA | 0.0388 | 0.0754 | 0.0461 | 0.9632 | 0.9880 |
| | Raw Depth | 0.0550 | 0.1458 | 0.0708 | 0.9179 | 0.9543 |
| L515
(D-Tof) | **Ours (CDM-L515)** | **0.0156** | **0.0297** | **0.0229** | **0.9754** | **0.9919** |
| | **Ours (CDM-D435)** | **0.0165** | **0.0349** | **0.0246** | **0.9613** | **0.9855** |
| | PromptDA*(515) | 0.0235 | 0.0666 | 0.0349 | 0.9291 | 0.9730 |
| | PromptDA*(435) | 0.0254 | 0.0438 | 0.0379 | 0.9234 | 0.9640 |
| | PromptDA | 0.0483 | 0.0400 | 0.0612 | 0.8867 | 0.9259 |
| | PriorDA | 0.5412 | 0.6134 | 0.9211 | 0.0850 | 0.1794 |
| | PromptDA | 0.0207 | 0.0515 | 0.0304 | 0.9480 | 0.9699 |
| | PriorDA | 0.0177 | 0.0385 | 0.0274 | 0.9502 | 0.9763 |
| | Raw Depth | 0.0312 | 0.0813 | 0.0475 | 0.9098 | 0.9429 |
| Helios
(I-Tof) | **Ours (CDM-L515)** | **0.0248** | **0.0403** | **0.0334** | **0.9468** | **0.9871** |
| | **Ours (CDM-D435)** | **0.0272** | **0.0457** | **0.0372** | **0.9297** | **0.9806** |
| | PromptDA | 0.0207 | 0.0515 | 0.0304 | 0.9480 | 0.9699 |
| | PriorDA | 0.0324 | 0.0597 | 0.0461 | 0.8984 | 0.9638 |
| | Raw Depth | 0.0384 | 0.0970 | 0.0542 | 0.8849 | 0.9578 |

Table 1: **Quantitative comparisons of metric depths prediction on Hammer (Jung et al., 2023) dataset (zero-shot evaluation).** The terms Filled. and Holed. refer to whether the low-quality depth is filled or directly given to the model for prediction. *(split) denotes fine-tuning on our synthesized datasets with the same augmentation strategy. Raw depth refers to the metric of directly using a low-quality depth image without a model. CDMs are named as the camera type, which are trained on the corresponding synthesized noise of that camera. All results are computed directly from the output of these models, without any alignment postprocessing.

evaluate the power of CDM and its benefits to robot manipulation, we develop a geometry-based sim-to-real pipeline that contains four main stages: scene construction, camera alignment, simulated data collection, and imitation learning. After, we directly deploy the trained policy onto real-world robots, with the corresponding CDM as an inference plug-in. In particular, we choose depth instead of pointclouds as the observation for the following reasons: 1) human relies on single-view stereo visual observations and can do many things; 2) pointclouds fused from multi-view cameras require careful camera calibrations and are more sensitive to irrelevant backgrounds. Details can be further referred to Appendix D.

## 4 EXPERIMENTS

The experiments involved are mainly threefold. 1) Does the camera depth model achieve better performance than other methods? 2) How does the accurate geometry information benefit real-world robot manipulation? 3) How the "sim-like" geometry contribute to zero-shot sim-to-real robotics manipulation? Note that our goal is not to identify if depth is a better visual modality than color, but to validate whether accurate geometry information contained in a more precise depth image can benefit manipulation. Therefore, the policies designed in our experiments are depth-only, excluding the effect of color information.

### 4.1 DEPTH PERFORMANCE

We mainly evaluate the trained camera depth models (CDMs) on the Hammer dataset (Jung et al., 2023), a real-world dataset that contains warped depth data paired with RGB images collected by three depth sensors: the RealSense D435 (stereo depth based on active structure light and IR images), L515 (a D-Tof camera), and a Lucid Helios (an I-Tof camera). Note that the dataset is not used for training, showing a zero-shot performance. We compared our CDMs against two baseline methods, PromptDA (Lin et al., 2025) and PriorDA (Wang et al., 2025c), both of which are metric depth prediction methods using prompt depth images and require hole-filling preprocessing during inference time. Since our CDM directly takes a prompt depth image as it is, we test two cases, *i.e.*, Filled and Holed, denoting whether the low-quality depth is filled or directly given to the model

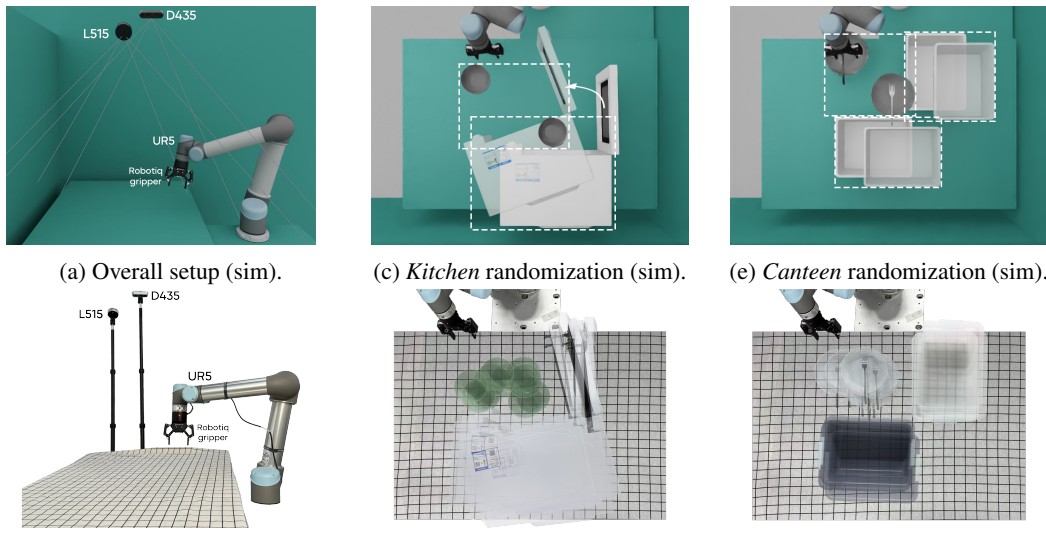

(a) Overall setup (sim).     (c) *Kitchen* randomization (sim).     (e) *Canteen* randomization (sim).

(b) Overall setup (real).     (d) *Kitchen* task distribution (real).     (f) *Canteen* task distribution (real).

Figure 3: **UR-5 manipulation experiment setup.** (a)(d) The overall setup for simulation and the real-world experiments, where we test two different cameras (D435 and L515) and train single-view policies for both cameras. In simulation, we add small randomization to the camera pose when generating demonstrations. (c)(e)The training environments of the kitchen and the canteen tasks in the simulation, where the dotted frame and the arrow denote the randomization range of objects. (d)(f) The test distributions of the two tasks, where we compose the position of the bowls and microwaves, following the randomization boundaries in the simulation.

for prediction. We also compared PromptDA fine-tuned on our synthesized dataset to show the advantage of the structure design.

The results are shown in Tab. 1, where we can observe and conclude several things. 1) Both PromptDA and PriorDA failed to obtain good depth prediction without hole-filling preprocessing, which requires additional computation time in real-time robot experiments. 2) Even with hole-filling, our CDMs achieve the state-of-the-art performance on corresponding data splits. 3) With the same training data and augmentation strategy, our CDMs still perform better than PromptDA, showing the advantage of our designed structure. 4) The model trained on specific synthesized camera noise data should work better on the same camera data split, like PromptDA; however, to our surprise, the CDM-L515 generalized well to the 435 datasets and can even achieve slightly better results than the CDM-D435 model, indicating the test cases of the D435 camera in the Hammer dataset can be generalized by the CDM-L515. 5) Both CDMs have better zero-shot generalization ability on the data split with a different depth sensor (the I-Tof Lucid Helios camera), potentially because CDMs solve some common noise problems among depth cameras.

## 4.2 IMITATION LEARNING WITH ONLY DEPTH

We aim to investigate how the accurate geometry information produced by the camera depth models benefits robot manipulation tasks. To this end, we design a pilot study that includes two pick-and-place tasks (*Toothpaste-and-Cup* and *Stack-Bowls*) using a daily-use depth camera, the RealSense D435. We manually collect 50 trajectories for each task through teleoperation of real robot and conduct the test at five different positions, with three trials at each position. In particular, for the *Stack-Bowls* task, we trained our policy on a normal-sized bowl and tested on bowls of five different sizes, including four unseen sizes. Without CDMs, the robot almost entirely failed in *Toothpaste-and-Cup* (0% success) and achieved only 20% success in *Stack-Bowls*. With CDM-D435, success improved markedly, reaching 40% in *Toothpaste-and-Cup*, and 60% in *Stack-Bowls*. Furthermore, the CDM-enabled policy generalized to bowls of unseen sizes, achieving consistent non-zero success across four novel sizes, while the baseline policy without CDMs failed completely. This highlights that accurate geometry not only boosts task performance but also enables robust generalization to previously unseen object scales. Detailed setup and results are provided in Appendix E.

Table 2: **Zero-shot sim-real results** using CDMs as the plugin in a real-world robot pipeline.

| Camera | Depth Model | Kitchen Task | | | | Canteen Task | | | | | |
|---|---|---|---|---|---|---|---|---|---|---|---|
| | | Pick Bowl | Put Bowl into Microwave | Close Microwave | Total | Pick Fork | Place Fork | Pick Plate | Dump Plate | Place Plate | Total |
| Sim (D435-View) | None | 43/50 | 33/50 | 32/50 | 30/50 | 40/50 | 28/50 | 47/50 | 45/50 | 33/50 | 21/50 |
| D435 | None | 0/30 | 0/30 | 0/30 | 0/30 | 0/30 | 0/30 | 0/30 | 0/30 | 0/30 | 0/30 |
| | PromptDA | 11/30 | 5/30 | 0/30 | 0/30 | 17/30 | 16/30 | 7/30 | 2/30 | 6/30 | 1/30 |
| | PriorDA | 16/30 | 8/30 | 7/30 | 7/30 | **30/30** | **30/30** | 1/30 | 0/30 | 0/30 | 0/30 |
| | CDM-D435 | **29/30** | **26/30** | **26/30** | **26/30** | **30/30** | **30/30** | **15/30** | **14/30** | **14/30** | **14/30** |
| | CDM-L515 | **29/30** | 22/30 | 16/30 | 14/30 | **30/30** | 29/30 | 0/30 | 0/30 | 0/30 | 0/30 |
| Sim (L515-View) | None | 43/50 | 34/50 | 37/50 | 32/50 | 40/50 | 26/50 | 46/50 | 43/50 | 31/50 | 20/50 |
| L515 | None | 0/30 | 0/30 | 0/30 | 0/30 | 0/30 | 0/30 | 0/30 | 0/30 | 0/30 | 0/30 |
| | PromptDA | 3/30 | 0/30 | 0/30 | 0/30 | 3/30 | 0/30 | 3/30 | 0/30 | 0/30 | 0/30 |
| | PriorDA | 17/30 | 3/30 | 2/30 | 2/30 | 10/30 | 8/30 | 3/30 | 3/30 | 3/30 | 3/30 |
| | CDM-D435 | 22/30 | 11/30 | 9/30 | 9/30 | 13/30 | 11/30 | 11/30 | 10/30 | 9/30 | 9/30 |
| | CDM-L515 | **25/30** | **18/30** | **18/30** | **18/30** | **24/30** | **24/30** | **22/30** | **22/30** | **22/30** | **22/30** |

## 4.3 ZERO-SHOT SIM-TO-REAL MANIPULATION

**Robot and task setup.** We construct our sim-to-real pipeline using a tabletop UR5 robot arm equipped with a Robotiq gripper, as illustrated in Fig. 3b. As mentioned before, the visual observation of the policy is the depth image from a single third-view camera. We design two long-horizon manipulation tasks: the kitchen task and the canteen task. 1) *The kitchen task* tests the ability to utilize articulation objects: the robot is required to pick up a bowl on the table, put it into the microwave, and then close the door of the microwave. Note that the microwave door is glass, which is seen as a hole from the original camera depth, and poses an additional challenge for depth capturing. 2) *The canteen task* requires recognizing and accurately grasping a slim fork and a thin plate, which are rather noisy, and the fork is even unseen from the original camera depth: the robot should pick up the fork and put it into the box in front of it, then pick up the plate and dump the trash into the left box, at last place the plate into the front box. We collect $\sim 680$ demo trajectories for the kitchen task and $\sim 800$ for the canteen task in simulation, train a policy by imitation learning, and directly deploy the policy in the real world by plugging in a depth model. The test setup is illustrated in Fig. 3d and Fig. 3f, where we test 10 positions and each for 3 times, resulting in 30 tests in total for both tasks. In the real world, we test two cameras: one is the RealSense D435 (IR Stereo) camera, and the RealSense L515 (D-Tof lidar) camera. For both cameras, we gather their intrinsics and calibrate the extrinsics using the method as mentioned in Section D.

**Results.** The zero-shot sim-to-real results are collected in Tab. 2, where we compare the policy performances using our CDMs against the same policy using two state-of-the-art prompt-based depth models, and directly using the raw depth image. We have several interesting observations: 1) The CDM works better under their specific camera type, although in the previous section, the CDMs showed generalization under the depth metric on a static dataset. 2) CDMs work better than previous baselines, even on different camera types, showing the advantage of the training dataset and the structure design. 3) The real-world policy performance matches the simulation performance, and some is even higher. The reason may be that the randomized position in the simulation is bigger, and some of them are difficult to complete all tasks.

**Total latency.** On a single RTX 4090 with a RealSense D435 providing the prompt depth, we measure end-to-end latency (pre-processing + Float32 model inference + post-processing) for different depth models: D3RoMa (Wei et al., 2024) (1.531±0.004s), PriorDA (0.154±0.005s), PromptDA (0.188±0.005s), and CDMs (0.151±0.002s). Without any additional engineering optimization or quantization, CDMs achieve the lowest latency, enabling control at over 6 Hz. In comparison, FoundationStereo on the Zed camera (limited in stereo RGB cameras) only achieves 3 Hz(0.319±0.002s).

## 5 CONCLUSION AND FUTURE WORKS

This work introduces camera depth models (CDMs), designed to provide high-quality geometric information by enhancing depth perception for specific depth cameras. By delivering accurate depth predictions, CDMs enable robust robotic manipulation in real-world settings with accurate geometry information, effectively bridging the sim-to-real geometry gap. Integrating CDMs with real-world depth cameras, we successfully transferred depth-only visuomotor policies, trained solely in sim-

ulation, to real robots on long-horizon manipulation tasks, achieving high success rates. These results underscore the critical role of accurate geometric information, provided by CDMs, in enabling generalizable and effective robotic manipulation. Although this study demonstrates CDMs in a depth-only imitation and sim-to-real pipeline, their potential extends far beyond this application. Future work could consider leveraging CDMs to relabel RGB-D data, enhancing policies with robust 3D representations. Moreover, by achieving simulation-level 3D perception in the real world and aligning sim-real geometry gaps, CDMs enable seamless integration of simulation and real-world 3D data. This approach could lead to more efficient data utilization strategies, fostering the development of large-scale robotic foundation models with human-level generalization ability for complex manipulation tasks.

## ACKNOWLEDGEMENTS

We would like to thank all members of the ByteDance Seed Robotics team for their support throughout this project. We also want to extend our gratitude to Tao Wang for his kind support of the depth camera devices and to Hang Li for his leadership of this team.

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

## A  DECLARATION OF LLM USAGE

Large Language Models (LLMs) were used in the preparation of this submission. Specifically, they assisted in editing and polishing the writing for grammar and clarity. All technical ideas, experimental designs, and results were developed by the authors. No LLMs were used to generate research content or experimental results.

## B  EXTENDED RELATED WORK FOR VISUAL SIM-TO-REAL

Sim-to-real transfer requires policies to overcome both the observation gap and the physics gap between simulated and real-world environments. Physics gaps, such as discrepancies in dynamics or friction, are often addressed by using domain randomization (Tobin et al., 2017; Peng et al., 2018), which trains policies to generalize across a range of physical parameters that encompass real-world variability. While this method is generally effective for locomotion tasks in legged robots (Xue et al., 2025; Kumar et al., 2021), manipulation tasks require more precise modeling because they depend heavily on accurate visual observations of objects, typically provided by RGB and/or depth images. Achieving robust sim-to-real policy transfer for manipulation tasks using RGB images requires high-fidelity simulation rendering to minimize the visual gap. Relying solely on simulator-generated images often demands extensive curriculum and augmentation design to ensure that the learned policy is effectively transferred to the real world (Wang et al., 2024a). Although advances in simulator rendering technologies (Xiang et al., 2020; NVIDIA, 2021) can help, recent real-to-sim approaches using neural rendering techniques demonstrate that reconstructing photorealistic scenes from real-world data can further reduce the visual gap (Han et al., 2025; Lou et al., 2024; Qureshi et al., 2024; Li et al., 2024b).

In addition to RGB images, some works utilize colorless 3D representations, such as point clouds and depth images, to reduce visual discrepancies between simulation and reality. For example, He et al. (2024) predicts the ray distances from simulated depth images to learn the reach-avoid value networks; Cheng et al. (2023), Zhuang et al. (2023), Zhuang et al. (2024), and Lai et al. (2024) employ depth images to train quadrupeds for collision avoidance and high dynamic locomotion; and Liu et al. (2024) uses depth images from two camera views for mobile manipulation tasks on a quadruped robot, which requires object segmentation to narrow the sim-to-real gap further. These approaches typically add noise and augmentations to simulated depth images and require post-processing of real-world depth data, such as clipping, hole filling, and temporal filtering, to address sensor imperfections. Alternatively, Hua et al. (2024) uses point clouds as visual input, but still adds noise in simulation and applies cropping and downsampling in the real world. Besides, Zhang et al. (2023) introduces a computation-cost method to simulate the depth with typical noise patterns rendered by a real-world stereo camera. However, simulating a real-world camera or adding noise in the simulation is a last resort, as it may deteriorate the rich geometry information and precise manipulation.

## C  DATASET

We design a multi-camera mount device to achieve highly efficient data collection, so that we can capture data simultaneously, as shown in Fig. 5. In detail, we mount seven cameras, including five RealSense cameras (D405, D415, D435, D455, L515), a ZED camera, and an Azure Kinect camera. For the ZED camera, we record the raw data and replay it with its 4 modes (performance, ultra, quality, neural) offline.

The collected dataset contains more than 17,000 images for each camera, sam- pled from videos at 5Hz, covering 7 different scenes, including kitchens, living rooms, markets, bedrooms, bathrooms, offices, and breakrooms, as shown in Fig. 4.

## D  SIM-TO-REAL MANIPULATION THROUGH CDMS

We describe the details of the four stages in our sim-to-real manipulation pipeline with CDMs.

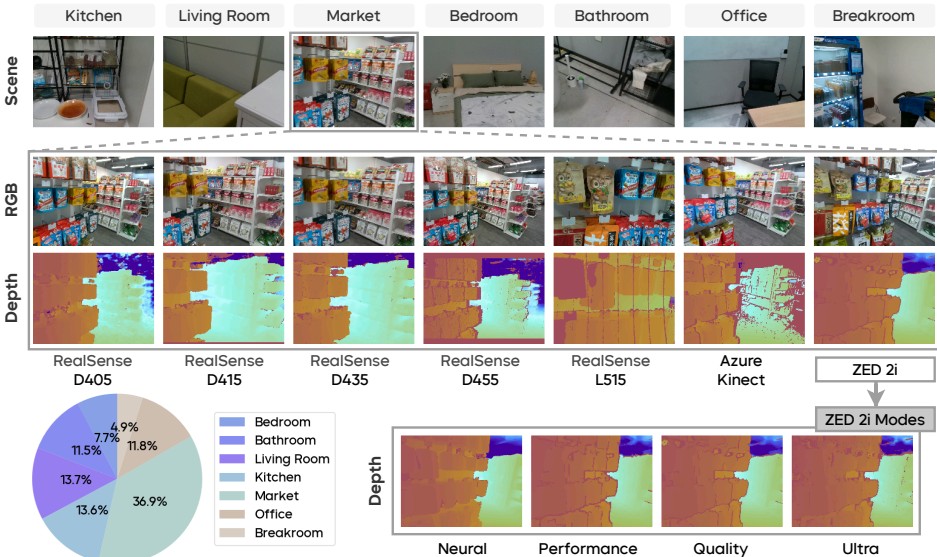

Figure 4: **Illustration of the collected ByteCameraDepth datasets**, which contains the raw depth data from 7 cameras, 10 modes (including 4 modes of the ZED 2i camera) in 7 different scenes.

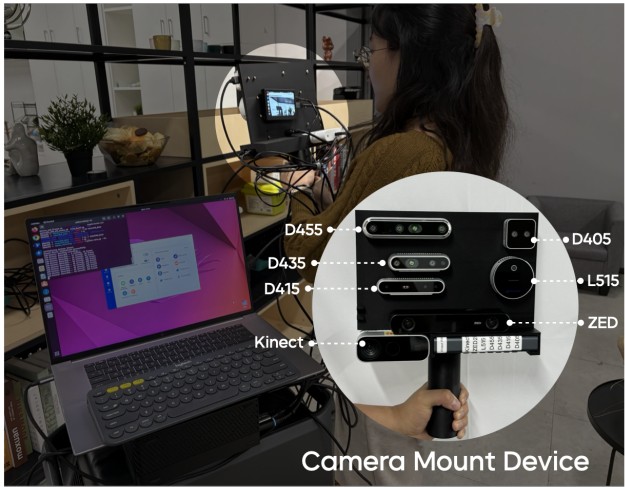

Figure 5: **Multi-camera mount device** for capturing the color-depth image pairs from multiple depth cameras all at once. In practice, we use two computers to capture all the data due to the USB bus bandwidth limits.

**Scene construction.** Since we rely on depth-only visual transfer, without any color information, the geometry-based sim-to-real pipeline does not require rigorous alignment of the exact appearance between objects and backgrounds in the simulation and the real world. Instead, we introduce geometrically similar objects and construct simple geometry as the background in the simulation. After building the environment, we set up the camera to obtain visual observations. We note that manipulation also requires a reasonable approximation of interactions between robots and objects, yet this is beyond the focus of this project. Thereafter, we manually assign the physical attributes of objects and modify the open-source robot description files to ensure plausible interactions, without aiming for absolute physical accuracy.

**Camera alignment.** To align the camera pose between the simulation and the real scene, we adopt a differentiable-rendering-based camera calibration method Chen et al. (2023) to estimate the cam-

era extrinsics in the real scene with minimal human effort, which only requires a few corresponding masks of the robot arm between real and virtual scenes. However, the calibrated poses are hard to perfectly align due to the differences in camera models between simulation and real-world sensors. To mitigate the gap from such misalignment, we slightly randomize camera poses during data collection in simulation, which helps the policy be robust to small discrepancies in viewpoint and better work in real-world deployments.

**Data generation.** To generate demonstrations efficiently in simulation, we develop an extension of MimicGen (Mandlekar et al., 2023), with whole-body control (WBC) (Haviland et al., 2022) to generate smoother, high-quality demonstrations. Our method introduces several data generation features like random reset to generate retrying demonstrations, controllable velocity for fine-adjustment, and so on. With WBC, the algorithm can be further extended to wheeled robots for mobile manipulation tasks. Details of the algorithm can be further referred to Appendix M.

**Imitation learning with depth.** We adopt a policy structure similar to the one used in Hua et al. (2024) and Lin et al. (2025). Although some previous works have shown some robustness using a point cloud based policy (Ze et al., 2024b;a; Hua et al., 2024), their point clouds are also transformed from the depth image captured by the depth camera and require cropping and downsampling to alleviate noises. Since now we can obtain an accurate depth where the geometry information is complete, we choose to use the depth image directly as the input to our policy. The policy encodes the depth image with a pre-trained ResNet, where the first layer of the network is replaced with a 1-channel convolutional layer; the proprioceptive states are encoded by a from-scratch MLP; a diffusion head (Chi et al., 2023; Ho et al., 2020) is adopted to predict the action sequence. We directly use the one-step single-view depth image rendered in simulation for training the policy, without adding any noise, but with only the `RandomShiftScaleRotate` augmentation to alleviate the camera calibration biases. Besides the depth image, the observation space also includes the joint position and the gripper status. In real-world deployment, we make our CDM a plugin between the camera and the policy, which predicts a clean depth image based on the raw depth image and the RGB image from the depth camera. The predicted depth image is then used for real-time policy inference.

## E    DEPTH-ONLY IMITATION LEARNING SETUP AND RESULTS

The real-world depth-only imitation experiments are conducted on a tabletop UR5 robot arm with a Robotiq gripper, using depth observations from a single RealSense D435 camera. The two pilot tasks are *Stack-Bowls* (pick bowl, stack bowl) and *Toothpaste-and-Cup* (pick toothpaste, put toothpaste into cup). The two tasks are illustrated in Figure 6, which depicts the success state of their subtasks. For each task, we collect 50 teleoperated demonstrations, and evaluation is performed across five different positions with three trials per position. In the *Stack-Bowls* task, the policy is trained on one normal-sized bowl and tested on five different sizes, including four unseen ones, to examine generalization.

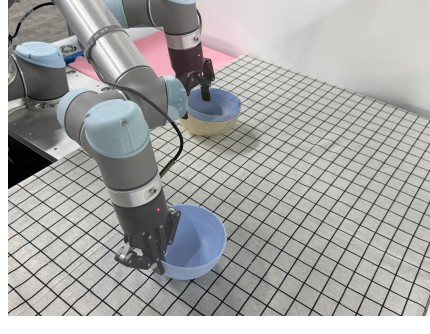
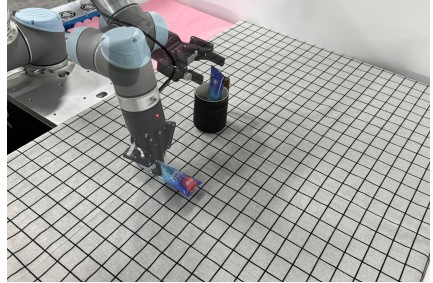

(a) *Stack-bowls* task.                (b) *Toothpaste-and-cup* task.

Figure 6: **The experiment setup of the depth-only imitation learning tasks.**

Results show that CDM-processed depth substantially improves success compared to raw depth (Table 3). Without CDMs, the robot almost entirely fails in *Toothpaste-and-Cup* and achieves low success in *Stack-Bowls*. With CDM-D435, success rates increase markedly across both tasks. Moreover, the CDM-enabled policy generalizes to bowls of unseen sizes, consistently achieving non-zero success across all four novel sizes, whereas the baseline without CDMs fails completely (Fig. 7).

Table 3: **Depth-only imitation results** with and without CDMs (15 trials per subtask; 50 demos per task for training).

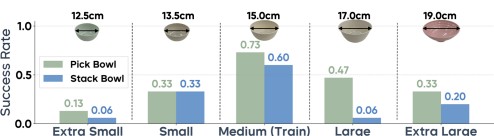

Figure 7: **Generalization over different bowl sizes.** With CDM, success remains non-zero across all unseen sizes; without CDM, success is zero.

| Depth Model | Toothpaste-and-Cup | | Stack-Bowls | |
|---|---|---|---|---|
| | Pick Toothpaste | Put Toothpaste into Cup | Pick Bowl | Stack Bowl |
| None | 0/15 | 0/15 | 6/15 | 3/15 |
| CDM-D435 | **10/15** | **6/15** | **11/15** | **9/15** |

# F  ADDITIONAL COMPARISONS WITH D3ROMA AND FOUNDATIONSTEREO

In this section, we provide additional comparisons between our CDMs, D3RoMa (Wei et al., 2024) and FoundationStereo (Wen et al., 2025). We report quantitative results on the Hammer (Jung et al., 2023) dataset only for D3RoMa, using the same splits and metrics as in Tab. 1. Note that FoundationStereo is specifically designed for stereo RGB cameras (e.g., ZED), and Hammer does not provide stereo RGB image pairs, so it cannot be evaluated quantitatively on this benchmark. Instead, we include qualitative comparisons with both D3RoMa and FoundationStereo on our ByteCameraDepth dataset samples to illustrate how different methods behave on real scenes.

## F.1  QUANTITATIVE COMPARISON WITH D3ROMA ON HAMMER DATASET

Tab. 4 reports the quantitative comparison between CDMs and D3RoMa on the D435, L515 and Helios splits of the Hammer dataset. We use exactly the same evaluation protocol as Tab. 1 in the main paper. Our CDMs consistently achieve better results than D3RoMa.

| Split | Method | L1 ↓ | RMSE ↓ | AbsRel ↓ | $\delta_{0.5}$ ↑ | $\delta_1$ ↑ |
|---|---|---|---|---|---|---|
| D435 (IR Stereo) | **Ours (CDM-D435)** | **0.0258** | **0.0404** | **0.0312** | **0.9842** | **0.9951** |
| | **Ours (CDM-L515)** | **0.0182** | **0.0338** | **0.0217** | **0.9877** | **0.9956** |
| | D3RoMa | 0.0338 | 0.0610 | 0.0432 | 0.9243 | 0.9740 |
| | Raw Depth | 0.0550 | 0.1458 | 0.0708 | 0.9179 | 0.9543 |
| L515 (D-ToF) | **Ours (CDM-L515)** | **0.0156** | **0.0297** | **0.0229** | **0.9754** | **0.9919** |
| | **Ours (CDM-D435)** | **0.0165** | **0.0349** | **0.0246** | **0.9613** | **0.9855** |
| | D3RoMa | 0.0282 | 0.0522 | 0.0428 | 0.9329 | 0.9652 |
| | Raw Depth | 0.0312 | 0.0813 | 0.0475 | 0.9098 | 0.9429 |
| Helios (I-ToF) | **Ours (CDM-L515)** | **0.0248** | **0.0403** | **0.0334** | **0.9468** | **0.9871** |
| | **Ours (CDM-D435)** | **0.0272** | **0.0457** | **0.0372** | **0.9297** | **0.9806** |
| | D3RoMa | 0.0400 | 0.0612 | 0.0523 | 0.8934 | 0.9690 |
| | Raw Depth | 0.0384 | 0.0970 | 0.0542 | 0.8849 | 0.9578 |

Table 4: **Additional comparison with D3RoMa on the Hammer dataset.** We report results on the three camera splits (D435, L515, Helios) using the same metrics and protocol as Tab. 1. Our CDMs achieve better geometric accuracy than D3RoMa on all splits.

## F.2  QUALITATIVE COMPARISONS ON BYTECAMERADEPTH DATASET

We further provide qualitative comparisons on our ByteCameraDepth dataset. Specifically, we visualize both depth maps and reconstructed point clouds for CDMs, D3RoMa and FoundationStereo,

together with the raw sensor outputs. Fig. 8 and Fig. 9 compare CDMs with D3RoMa. The former shows depth map comparisons, while the latter shows the corresponding reconstructed point clouds. Similarly, Fig. 10 and Fig. 11 compare CDMs with FoundationStereo in terms of depth maps and point clouds, respectively.

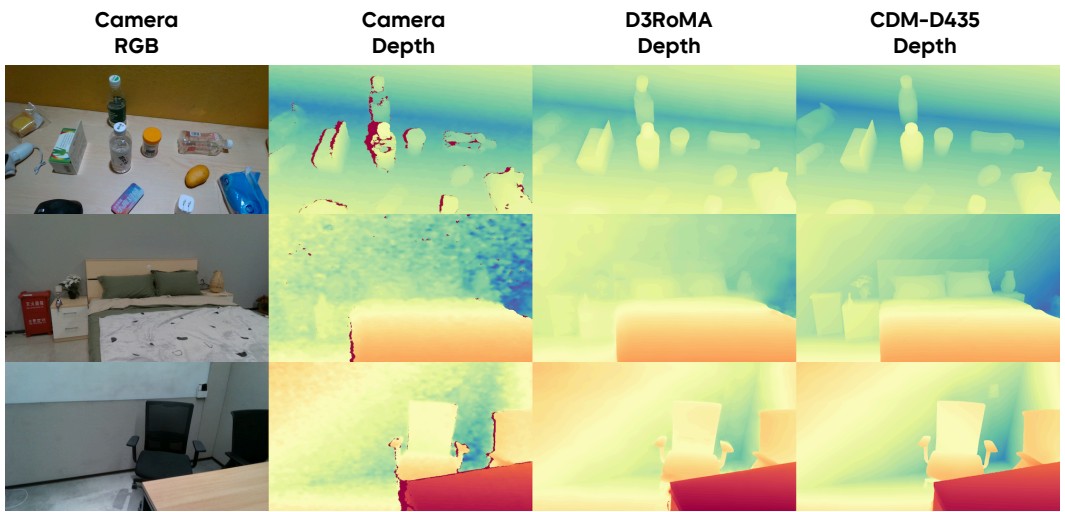

Figure 8: **Depth prediction comparison between CDMs and D3RoMa.**

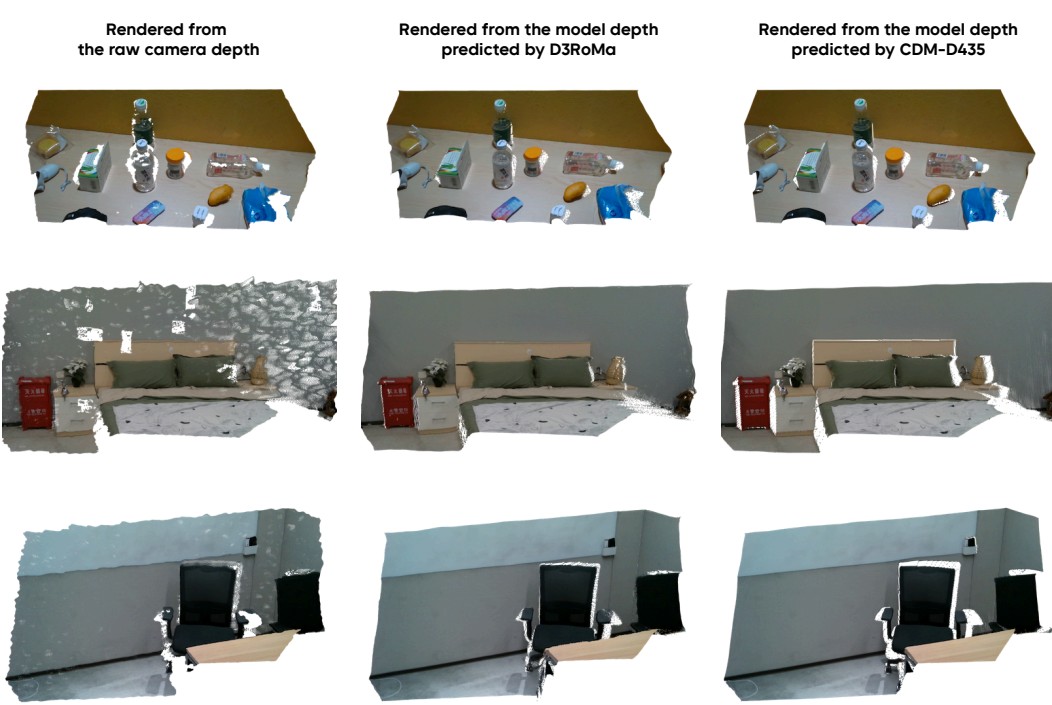

Figure 9: **Reconstructed point cloud comparison between CDMs and D3RoMa.**

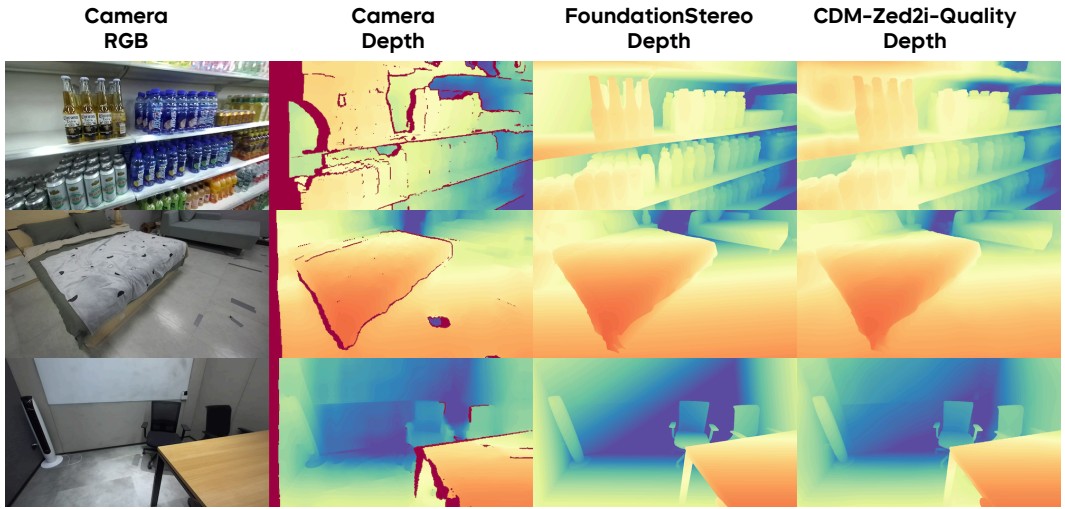

Figure 10: **Depth prediction comparison between CDMs and FoundationStereo.**

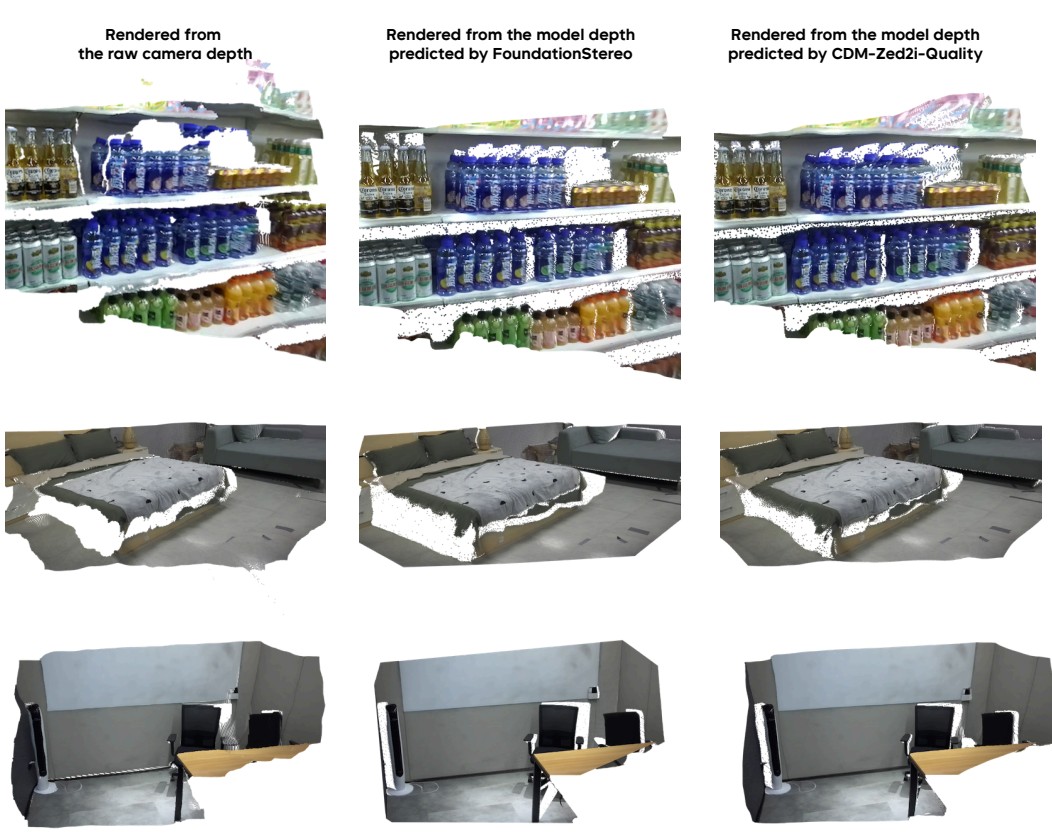

Figure 11: **Reconstructed point cloud comparison between CDMs and FoundationStereo.**

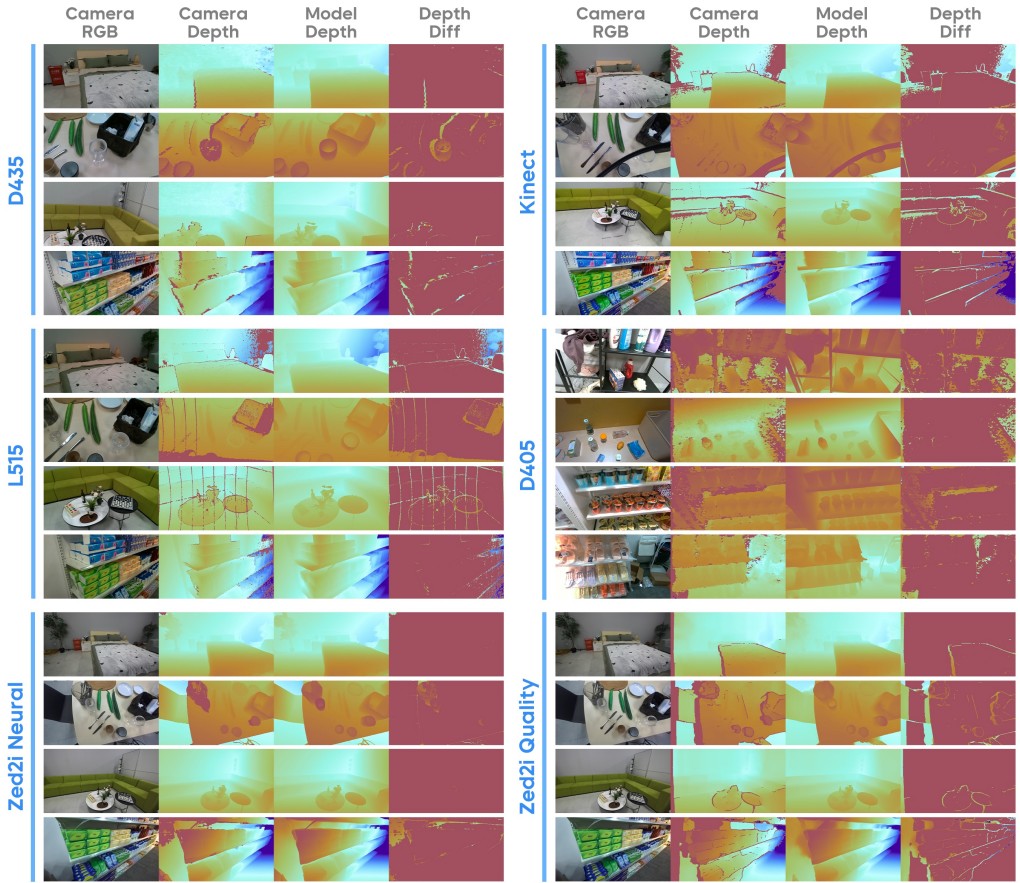

Figure 12: **Depth predictions visualization of extended camera depth models,** including CDM-D435, CDM-D405, CDM-L515, CDM-Kinect, and CDM-Zed2i (Neural & Quality mode) on their corresponding camera captures. Note that the depth difference is visualized in a range of 0-0.2m.

## G EXTENDED CAMERA DEPTH MODELS

Except for the two daily-used camera depth models (CDMs) used in the robot manipulation experiments, *i.e.*, RealSense D435 and RealSense L515, we further train three CDMs for RealSense D405, Azure Kinect, and Zed2i (Neural mode). Since we do not have the corresponding real-world dataset with the ground-truth depth label to quantitatively evaluate them, we simply visualize their predictions on representative scenes from the ByteCameraDepth dataset, shown in Fig. 12. Since RealSense D435 uses a similar depth technology to RealSense D415 and RealSense D455, they share a similar noise mode, so we visualize CDM-D435 predictions upon these two cameras and find that CDM-D435 can also provide good predictions. We open all model weights to allow the community to further test and improve them.

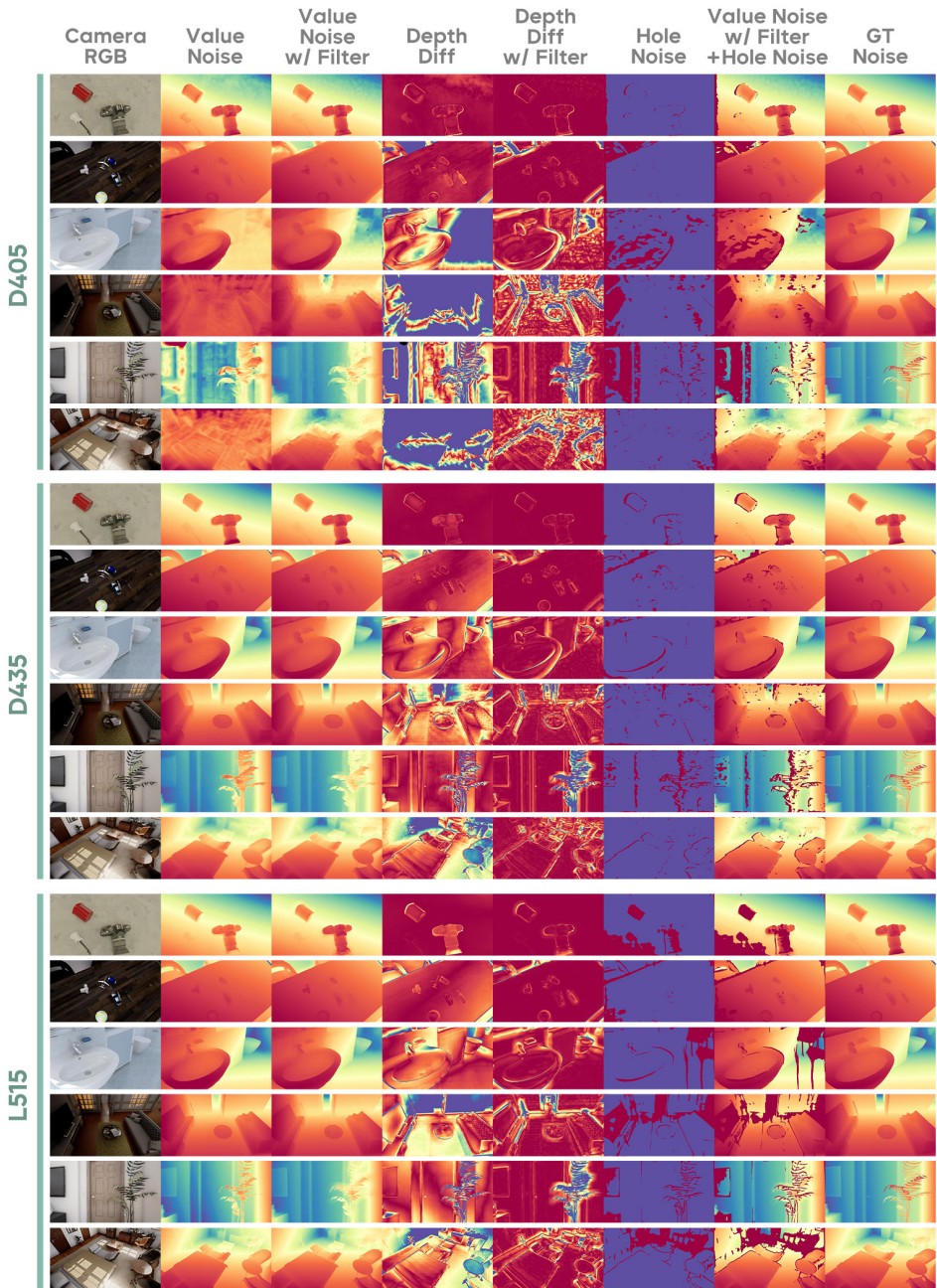

Figure 13: **Synthesized noise of RealSense D405, D435 and L515.**

## H    VISUALIZATION OF SYNTHESIZED NOISE

We illustrate the synthesized noise samples from the simulation datasets in Fig. 13 and Fig. 14, produced by the noise models, including the hole noise and the value noise, learned from the collected ByteCameraDepth dataset. We observe that the noise models learn typical noise patterns of the depth camera, *e.g.*, the failures on transparent objects. Additionally, the guided filter fills the scale gap between the value noise and the ground truth.

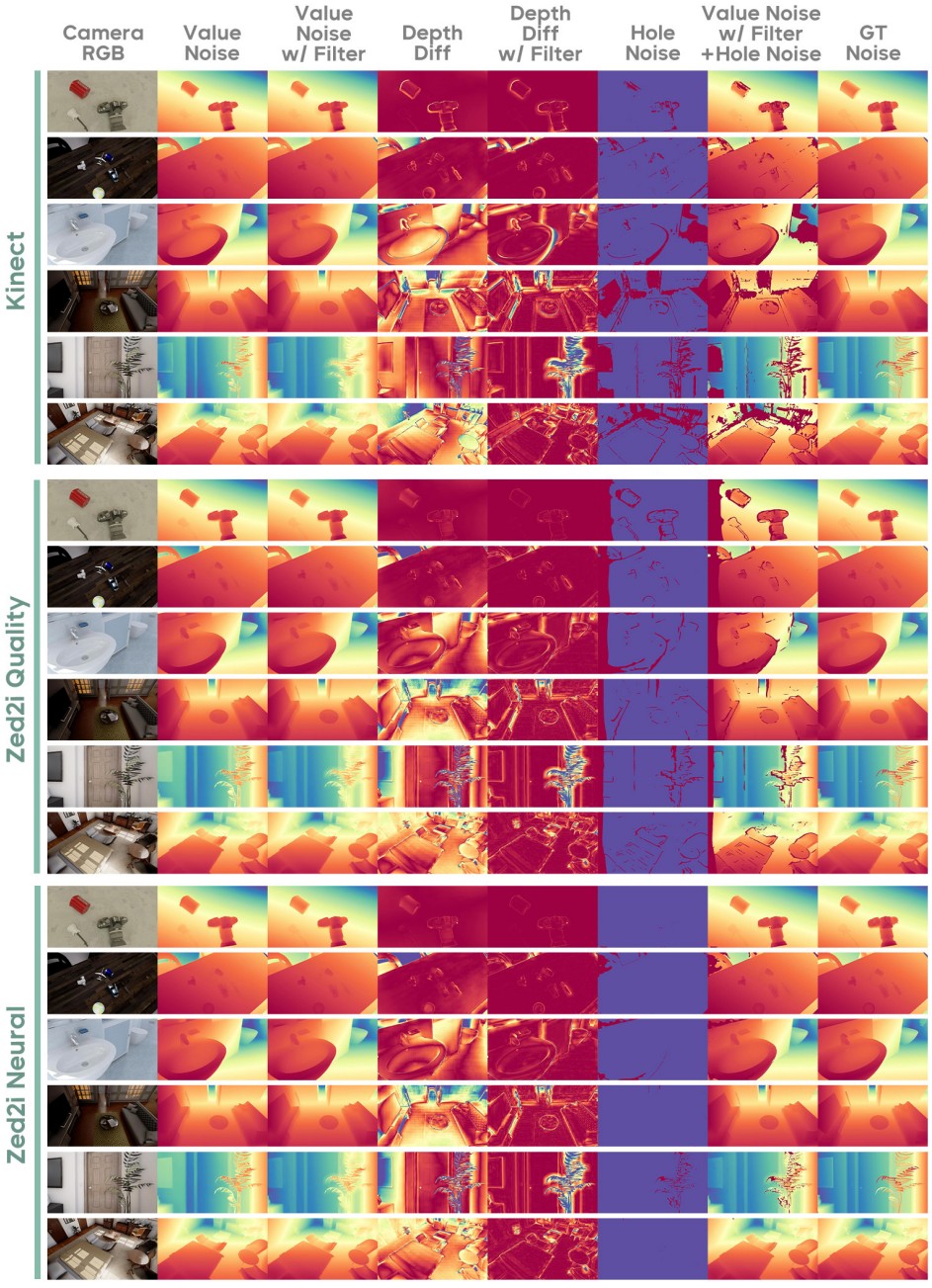

Figure 14: **Synthesized noise of Azure Kinect and two modes of ZED2i.**

# I  DEPTH ACCURACY W.R.T DISTANCE

Before leaving the factory and being sold to customers, a depth camera will be evaluated at various distances to fully examine its desired working range. To understand the work range of CDMs and as a reference to help people use them, we also provide the depth accuracy w.r.t the distance of CDMs (CDM-D435 and CDM-L515) on the Hammer dataset, in terms of absolute and relative error and the L1 error, shown in Fig. 15. From the accuracy curves, we observe that the raw depth has a larger error than the camera producer claims, for example, the RealSense D435 should have a less than 2% error rate when working under 1∼2 meters, which may be the bias of the dataset. Upon this dataset, CDMs can achieve a high accuracy, whose trend follows the accuracy of the original prompted depth.

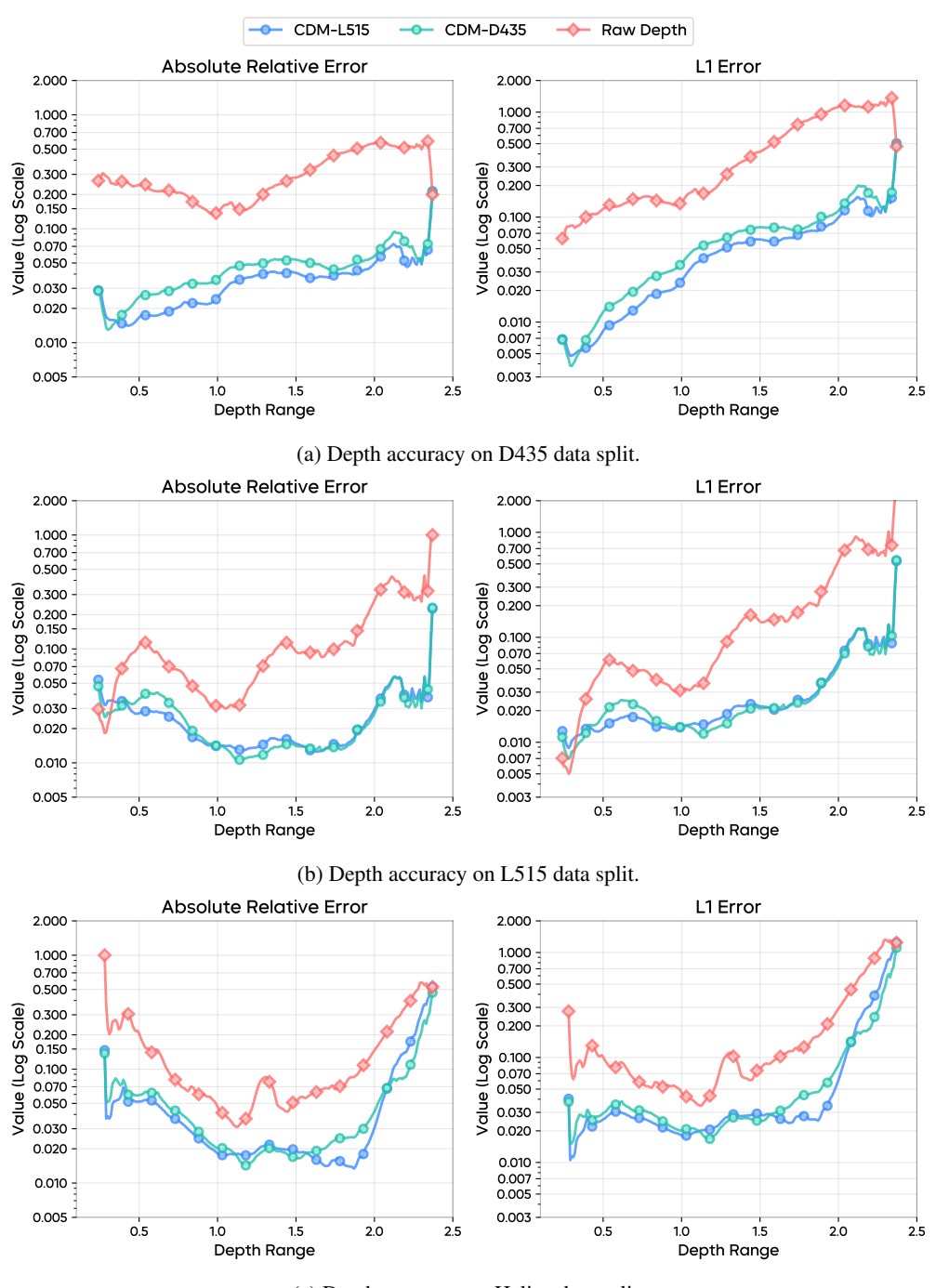

(a) Depth accuracy on D435 data split.

(b) Depth accuracy on L515 data split.

(c) Depth accuracy on Helios data split.

Figure 15: **Depth accuracy evaluation of CDMs and other models on the Hammer dataset.**

## J  DEPTH COMPARISON

We provide a detailed visual comparison between the raw camera depth and the predicted depth by the proposed CDMs, shown in Fig. 16. We can easily observe that both of these two representative depth cameras have their typical noise and failure modes. For example, both cameras fail to recognize the glass of the microwave and the metal fork; D435 has noisy depth on the plaid tablecloth; L515 has problems with the reflective outside part of the microwave, and the gripper fingers. In comparison, CDMs can provide accurate and complete geometry information.

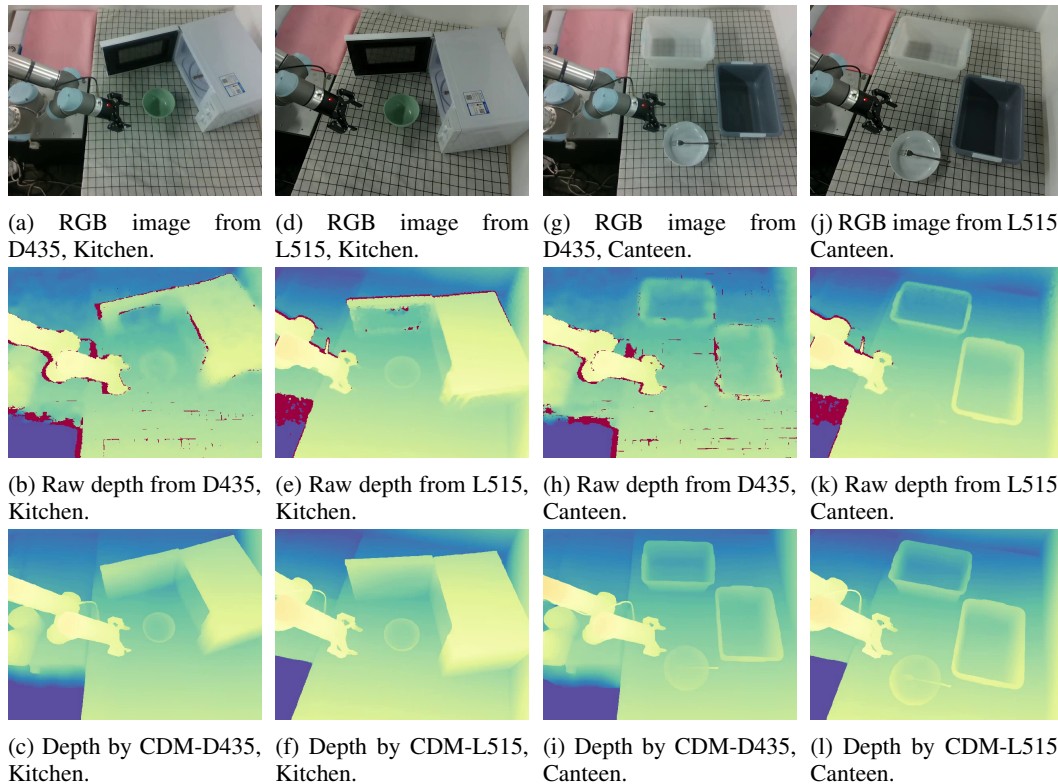

(a) RGB image from D435, Kitchen.
(d) RGB image from L515, Kitchen.
(g) RGB image from D435, Canteen.
(j) RGB image from L515, Canteen.

(b) Raw depth from D435, Kitchen.
(e) Raw depth from L515, Kitchen.
(h) Raw depth from D435, Canteen.
(k) Raw depth from L515, Canteen.

(c) Depth by CDM-D435, Kitchen.
(f) Depth by CDM-L515, Kitchen.
(i) Depth by CDM-D435, Canteen.
(l) Depth by CDM-L515, Canteen.

Figure 16: **Detailed real-world cases of two representative depth cameras,** RealSense D435 (active IR stereo camera) and RealSense L515 (lidar camera), including color images (first row), camera depth images (second row), and depth predicted by camera depth models proposed in this project (third row).

## K    RENDERED POINT CLOUD COMPARISON

We further compared the rendered point clouds transformed from the raw camera depth and the ones predicted by the camera depth models (CDMs), shown in fig:pcd-real (two real-world imitation tasks), Fig. 18, and Fig. 19 (two sim-to-real tasks). From the rendering results, we can easily observe that the point clouds rendered by the raw depth camera are much noisier, where the objects are distorted and convey wrong geometry information. In comparison, the CDM provides a clean point cloud where the objects maintain most of their original geometry.  It is worth noting that in the Canteen task, the geometry of the fork from the raw camera depth is integrated within the plate; the one predicted by the CDM is better, but still inaccurate. This is because the raw camera depth does not provide any useful information about the fork, and the model has to predict the whole from the semantic information of the color image, which may be confusing. We encourage readers to further visit the project page for more interactive point cloud rendering demos.

Rendered from the raw camera depth

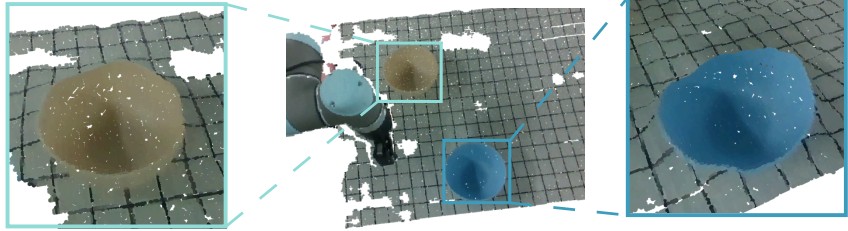

Rendered from the model depth predicted by CDM-D435

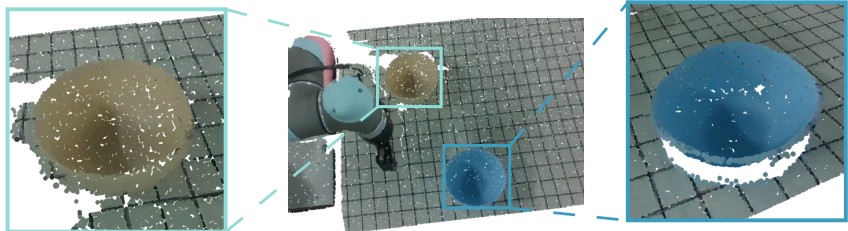

(a) Point cloud comparison of the Stack Bowl task.

Rendered from the raw camera depth

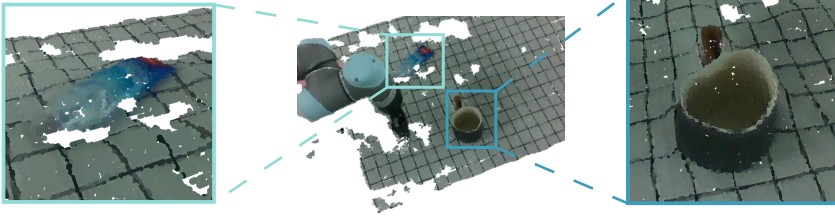

Rendered from the model depth predicted by CDM-D435

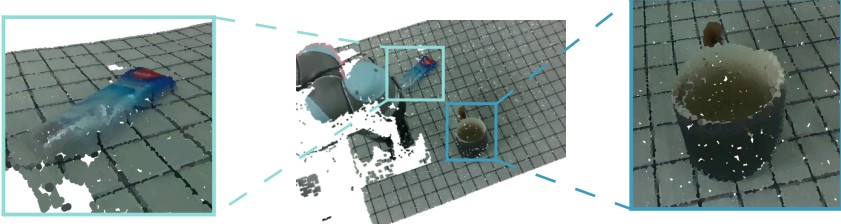

(b) Point cloud comparison of the Toothpaste task.

Figure 17: **Rendered point cloud comparison on two real-world imitation tasks** between the raw camera depth and the predicted depth of CDM-D435, upon the RealSense D435 camera.

Rendered from the raw camera depth

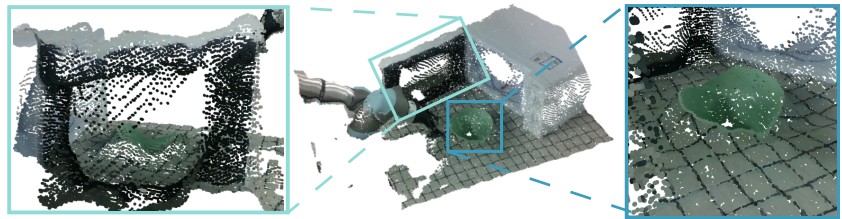

Rendered from the model depth predicted by CDM-D435

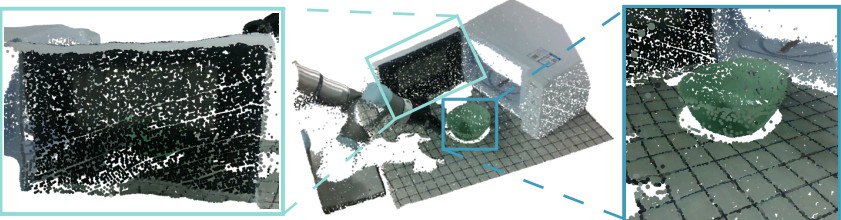

(a) Point cloud comparison of the Kitchen task.

Rendered from the raw camera depth

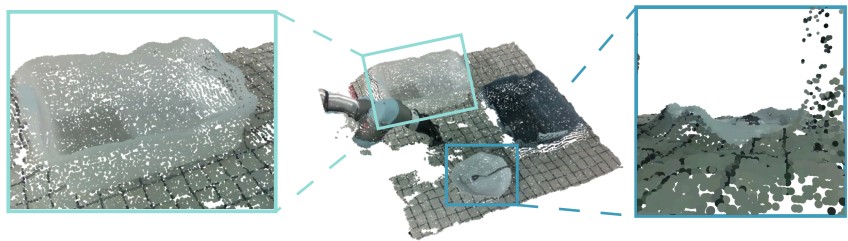

Rendered from the model depth predicted by CDM-D435

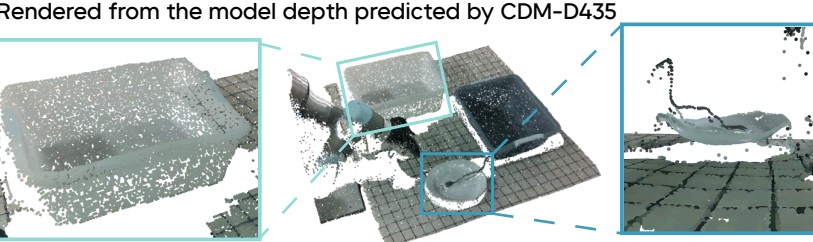

(b) Point cloud comparison of the Canteen task.

Figure 18: **Rendered point cloud comparison on sim-to-real tasks** between the raw camera depth and the predicted depth of CDM-D435, upon the RealSense D435 camera.

Rendered from the raw camera depth

Rendered from the model depth predicted by CDM-L515

(a) Point cloud comparison of the Kitchen task.

Rendered from the raw camera depth

Rendered from the model depth predicted by CDM-L515

(b) Point cloud comparison of the Canteen task.

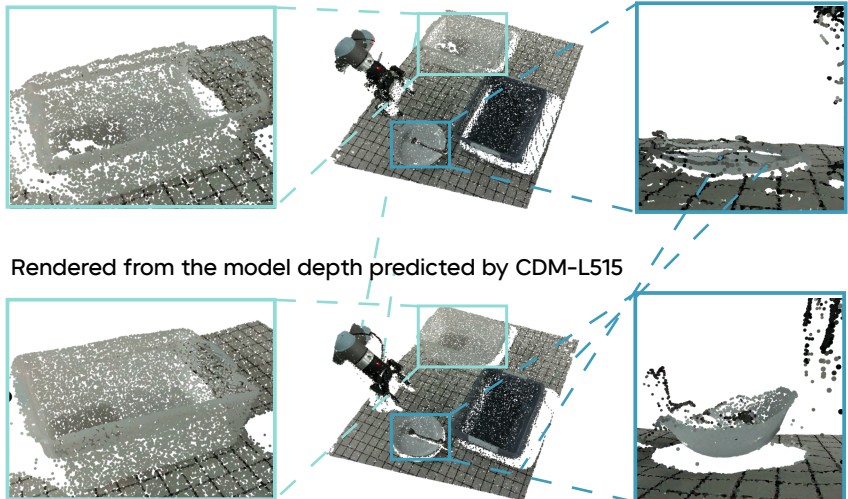

Figure 19: **Rendered point cloud comparison on sim-to-real tasks** between the raw camera depth and the predicted depth of CDM-L515, upon the RealSense L515 camera.

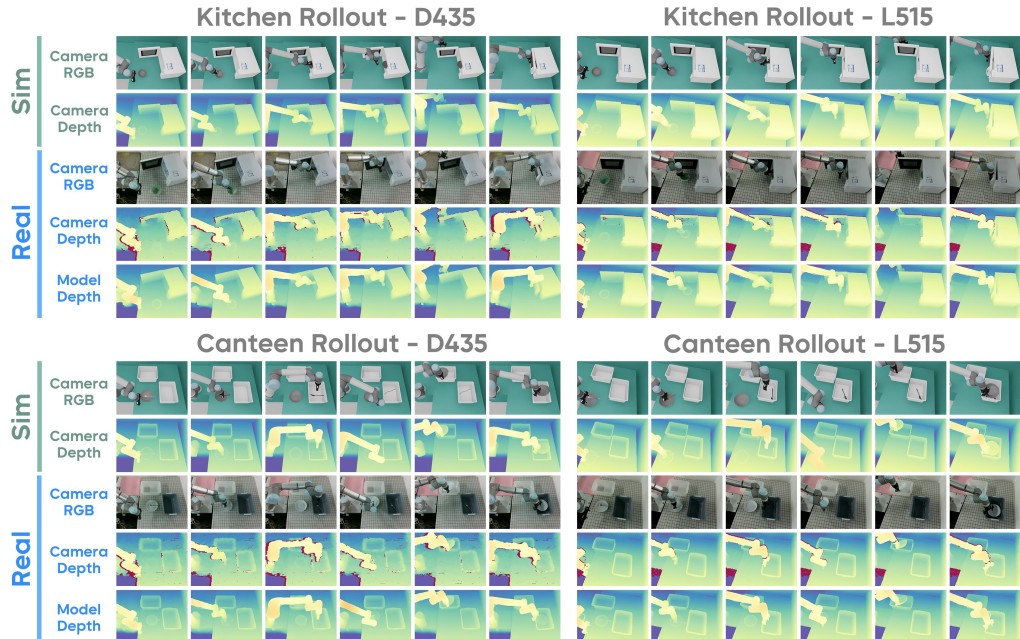

Figure 20: **Sim-Real rollouts comparison on two tasks in the sim-to-real experiments,** including two views (D435 view and L515 view). For the simulation, we show the rendered RGB and depth images; as for the real experiments, we visualize the RGB images and the depth images from the depth camera, with the predicted depth from the corresponding camera depth model.

## L COMPARISON OF SIM-REAL ROLLOUTS

We visualize the policy rollouts on two tasks in the sim-to-real experiments in Fig. 20, where we compare the key frames from the simulation and the real world separately. It is readily apparent that the camera depth model provides high-quality, simulation-like depth, offering accurate geometry information in the real world and thus bridging the geometry gap between simulation and reality.

# M  DATA GENERATION WITH WBCMIMICGEN

## M.1  ALGORITHM

To generate demonstrations efficiently in simulation, inspired by Haviland et al. (2022), we propose WBCMimicGen, which extends with whole-body control (WBC). Compared to the original MimicGen algorithm, which utilizes linear interpolation of end-effector poses to generate trajectories, WBCMimicGen optimizes target joint velocities with WBC, considering the manipulability, joint limits, joint velocity limits together in the QP problem and thereby generating smoother, high-quality demonstrations. This approach can be further extended to wheeled robots for mobile manipulation tasks. In the simulation, the advantage of precise perception, without considering any error, helps utilize classical control methods, such as WBC, and thus we can get more safe, smooth and controllable trajectories.

Specifically, we regard the data generation problem as solving a trajectory of whole-body joint velocities that enables the end-effector to move with a specific velocity. Formally, denote joint velocities as $x$, this problem can be modeled as a quadratic programming (QP) problem (Haviland et al., 2022):

$$\min_{x} \quad f_o(x) = \frac{1}{2}x^\top Q x + \mathcal{C}^\top x \tag{10}$$
$$\text{subject to } \mathcal{J}x = {}^b\nu_e \,,$$
$$\mathcal{A}x \leq \mathcal{B} \,,$$
$$\mathcal{X}^- \leq x \leq \mathcal{X}^+ \,,$$

where $x = (a_{\text{base}}, \dot{q}_{\text{active}}, \delta_1, \delta_2, \cdots, \delta_i)$ and $\mathcal{X}^{+,-}$ is the limits; $a_{\text{base}}$ is the velocities of the robot base; $\dot{q}_{\text{active}}$ is the velocity of the joints related to the end-effectors in the QP (so called the active joints); $\delta_i$ are slack variables that can help construct a solvable QP. Without loss of generality, suppose there are $k$ end-effectors and $n$ joints, these variables can be expressed as:

$$
\begin{aligned}
\mathcal{Q} &= \text{diag}(\lambda_q, \lambda_{\delta_1}, \cdots, \lambda_{\delta_k}) \in \mathbb{R}^{(n+6k)} \,, \\
\mathcal{C} &= \begin{pmatrix} \hat{\mathbf{J}}_m + \epsilon \\ 0_{6k \times 1} \end{pmatrix} \in \mathbb{R}^{(n+6k)} \,, \\
\mathcal{A} &= (\mathbf{1}_{n \times (n+6k)}) \in \mathbb{R}^{n \times (n+6k)} \,, \\
\mathcal{B} &= \begin{pmatrix} 0_b \\ \eta\frac{\rho_0 - \rho_s}{\rho_i - \rho_s} \\ \vdots \\ \eta\frac{\rho_n - \rho_s}{\rho_i - \rho_s} \end{pmatrix} \in \mathbb{R}^n \,.
\end{aligned}
\tag{11}
$$

Here $\hat{\mathbf{J}}_m$ is the manipulability Jacobian, $\epsilon$ is the base to end-effector angle, and $\rho$ is the distance to the nearest joint limit, encouraging the joint not to stay too close to the limit.

## M.2  COMPARISON RESULTS

To evaluate the data quality generated by WBCMimicGen, we compare the trajectory smoothness, measured by the mean absolute acceleration and the root mean square (RMS) jerk (*i.e.*, the averaged rate of acceleration change) metrics against the data generated by the original MimicGen Mandlekar et al. (2023) algorithm. Previous works Gasparetto and Zanotto (2007; 2008) use metrics like these as objectives for better smoothness. As shown in Tab. 5, WBCMimicGen consistently generates smoother trajectories with significantly lower acceleration and jerk values across all joints. This improvement stems from the quadratic programming formulation of WBC, which incorporates velocity regularization and enforces joint velocity limits. We encourage readers to further visit the project page for a direct visual comparison of the generated trajectories.

Simulation experiments further validate these improvements. As detailed in Tab. 5, models trained on WBCMimicGen data achieve higher success rates (72% vs 56% for Kitchen, 42% vs 24% for Canteen) while maintaining substantially lower RMS jerk and acceleration. The baseline approach

Table 5: **Comparison of the generated demonstrations over every robot joint (UR5).**

| Task | Method | Joint 1 | Joint 2 | Joint 3 | Joint 4 | Joint 5 | Joint 6 |
|---|---|---|---|---|---|---|---|
| | | **Mean absolute acceleration (rad/s$^2$) ↓** | | | | | |
| Kitchen | MimicGen | 1.284 | 1.253 | 0.910 | 8.115 | 4.457 | 6.702 |
| | WBCMimicGen | **0.183** | **0.495** | **0.311** | **4.925** | **2.810** | **6.539** |
| Canteen | MimicGen | 1.180 | 1.792 | 1.310 | 6.487 | 2.961 | 1.177 |
| | WBCMimicGen | **0.028** | **0.064** | **0.080** | **2.200** | **0.433** | **0.709** |
| | | **RMS jerk (rad/s$^3$) ↓** | | | | | |
| Kitchen | MimicGen | 461.374 | 313.714 | 277.185 | 1047.850 | 852.199 | 1252.070 |
| | WBCMimicGen | **58.774** | **112.983** | **84.063** | **767.230** | **516.074** | **1020.545** |
| Canteen | MimicGen | 435.487 | 384.092 | 425.695 | 1004.206 | 592.221 | 282.824 |
| | WBCMimicGen | **8.610** | **19.879** | **25.812** | **547.281** | **111.007** | **207.548** |

exhibits much larger acceleration at action chunk boundaries, which increases the likelihood of dropping objects. In contrast, WBCMimicGen's smoother trajectories enhance both task reliability and safety, making them more suitable for real-world deployment.

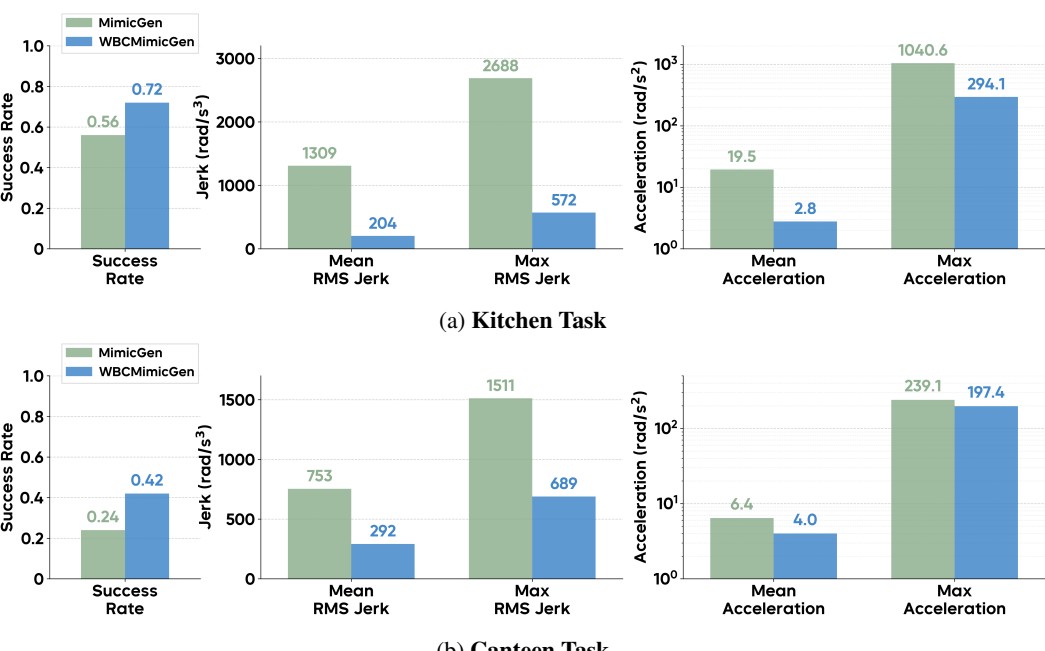

Figure 21: **Performance comparison between policy models trained on demonstrations** generated by MimicGen and our WBCMimicGen, on (a) Kitchen Task and (b) Canteen Task. The policy learned from WBCMimicGen reflects a higher success rate while keeping smooth on the rollout policy trajectory (lower RMS jerk and acceleration indicate).

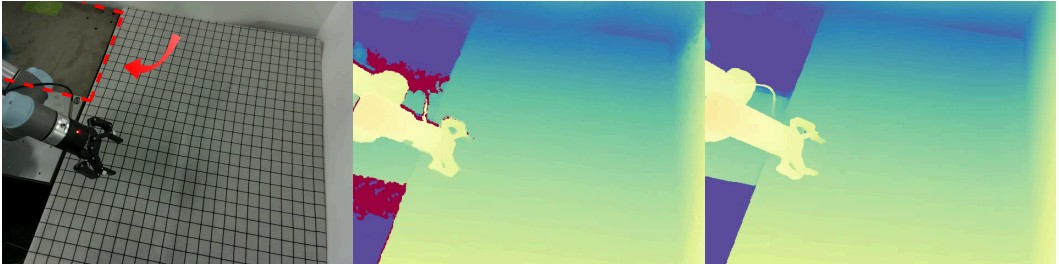

Figure 22: **A typical failure case of CDM-L515,** caused by the wrong prompt and the less informative semantic information contained in the RGB image.

## N  LIMITATIONS AND FAILURE CASES

Although CDMs can fix many errors of the source depth cameras due to the semantic information in the RGB image, they are still easy to be effected by the prompted depth image and fall into some failure cases when the monocular semantic information is not enough to fix the error. In other words, when the prompted camera depth image has wrong metrics for a large region, the predicted depth can be misled. Here we provide a typical case on CDM-L515 in Fig. 22, where the red dashed line highlights the area where the prompted camera depth hints that it is a hole. This is due to the metal plane causes the failure perception for the RealSense L515 is a LiDAR depth camera. Additionally, the RGB image does not bring informative semantic information for the CDM to fix that error.

