# OpenReview forum: "Manipulation as in Simulation: Enabling Accurate Geometry Perception in Robots"
_ICLR.cc/2026/Conference — ICLR 2026 Poster_

### Official Review · Reviewer_b9og · 2025-10-27

**Soundness:** 2
**Presentation:** 3
**Contribution:** 2
**Rating:** 4
**Confidence:** 4

**Summary:**

This paper addresses the challenge of transferring robotic manipulation policies from simulation to the real world, identifying the noise and inaccuracies of real-world depth sensors as a primary obstacle.The authors propose "Camera Depth Models (CDMs)," a plug-in module designed to take raw RGB and noisy depth signals from a specific camera and output a clean, accurate, "simulation-like" metric depth map.

**Strengths:**

1. The paper tackles a highly relevant and practical problem in robotics: the sensor-reality gap that plagues sim-to-real transfer. The proposed approach of explicitly de-noising the real-world perception to match the clean simulation environment, rather than the other way around, is a clear and interesting contribution.
2. The introduction of the ByteCameraDepth dataset, which captures data from 7 different depth cameras across 10 modes and multiple scenes, is a valuable contribution to the community for studying and modeling depth sensor noise

**Weaknesses:**

1. **Missing Critical Baseline:** The paper's primary motivation is that depth noise breaks sim-to-real transfer. The proposed solution is to clean the real-world depth (Sim-Clean -> Real-Clean). However, the paper fails to compare this against the most obvious and widely-used baseline: making the policy robust to noise via domain randomization (i.e., Sim-Noisy -> Real-Noisy). It is common practice to train policies on simulated data with added noise patterns precisely to make them robust to real-world sensor imperfections. The paper only compares its depth *model* (CDM) against other depth *models* (e.g., PromptDA) but does not compare its *manipulation pipeline* against this standard sim-to-real baseline. This is a significant omission that weakens the central claim.
2. **Questionable Premise:** Related to the first point, the premise that policies *cannot* be trained on noisy data is debatable. While the paper shows that a policy trained on *clean* data fails on *noisy* data (Table 2, "None" model), this is expected. It does not prove that a policy trained on noisy data would also fail. Many existing works successfully train policies on real-world, noisy data (for imitation learning) or on noise-augmented simulation data (for sim-to-real), as the policy learns to filter or ignore the noise implicitly. The paper overstates the necessity of perfectly clean depth for manipulation.
3.  **Lack of Robustness Analysis:** For a paper focused on real-world manipulation, there is no analysis of the policy's robustness. The experiments are run in a controlled setting. There are no ablations involving physical perturbations, dynamic clutter, or other disturbances during policy execution. It is possible that the policy's success is brittle and highly dependent on the (now clean) geometric information, but lacks the robustness that might be learned from training on more varied (i.e., noisy) data.

**Questions:**

1. The core of the paper focuses on the CDM models. The actual manipulation policy, which is critical to the paper's claims, is barely discussed in the main text. Section 4.3 briefly mentions "imitation learning" , but the specific architecture (a ResNet encoder with a diffusion head) is relegated to Appendix D. This makes the manipulation aspect of the paper feel like an afterthought and difficult to evaluate properly.
2. The paper's title and abstract frame it as a manipulation paper, but the experimental validation for manipulation is limited. The zero-shot evaluation rests on two tasks (Kitchen and Canteen) presented in one table (Table 2). While the tasks are well-chosen, this is a small number from which to conclude that this method "enables generalizable and effective robotic manipulation"[L 880].

---

> ### Author Response · Authors · 2025-11-21
>
> Thanks for your feedback.
>
> **Regarding weakness**
> 1. Our goal is not “noisy block sim2real”, but to exploit the fact that in the real world, we can now obtain much cleaner geometry. In addition, we provide **sufficient evidence** in Sec. 4.2 with real-data imitation to show how policy-trained clear geometry information from CDM-denoised depth outperforms noisy depth. With that proven, there is less need to show how depth with noise augmentation trained in sim transfers in real. Therefore, Sec 4.3 focuses on showing a policy trained without any noise augmentation in sim works well in reality with CDM. Besides, we want to highlight that the previous necessity of randomization is because real-world depth perception is bad, and we do not have to make such a concession as we get better real-world perception.
> 2. - We do **NOT** claim policy cannot be trained on noisy data, but **highlight** how accurate geometry information (clean depth) helps manipulation.
>    - We have included experiments (Sec 4.2) to show the advantage of clean depth by training a depth-only policy, and demonstrate the generalization across similar geometric shapes (Table 3, Figure 7 in Appendix E).
>    - We compared the way of dealing with depth/pointclouds of current methods in robotic manipulation in both Section 2 and Appendix B. We want to highlight that most of the current usage in simulation, in fact, damages the accurate 3D information (such as noise, destructive augmentation, and physics-grounded depth sensor simulation). In reality, they need to filter the perceptual data (like downsample and crop). Such a concession has to be made because real-world depth perception is quite bad. As we argued in 2), accurate depth helps the policy in learning geometry-related features, and also fills sim-to-real geometry gaps. Therefore, having such a plug-in-style tool for manipulation has only benefits when the computation cost is not a bottleneck.
> 3. We demonstrate positional generalization of the learned policy in both Sec 4.2 and Section 4.3 (Figure 3). Our policy also learned retry behaviors and dynamic disturbance as we collected such data (as shown in the supplemented videos). In short, we are not making the policy overfit to the clean depth, but by providing noiseless geometry information to improve the policy’s generalization ability through geometry.
>
> **Regarding questions**
> 1. - **The core motivation of this paper is to solve the bad depth perception problem in the robotic manipulation pipeline**, which came up after our disappointing experience with the current widely-used depth camera to obtain 3D information in depth-related manipulation projects.
>    - Therefore, the manipulation experiments are designed to identify the proposed CDMs’ benefits for manipulation tasks.
>    - We clearly claim our contribution in Sec 1. Due to the page limit, we cannot include all details in the main text; specifically, the architecture-related information you mentioned is off-the-shelf and lacks originality.
> Therefore, we sincerely ask you to evaluate the property of this work from the contribution we mentioned.
> 2. - As mentioned above, the manipulation experiments are designed to identify the proposed CDMs’ benefits for manipulation tasks.
>    - We included **2 real-world imitation learning tasks** (Sec 4.2), and **2 long-horizon** sim-real tasks (Sec 4.3). Compared with many existing manipulation papers that only contain multiple **single-stage tasks**, our long-horizon tasks contain **7 sub-tasks** in total. We reported both the independent results of each sub-task and their consecutive results in total to show the generalization and effectiveness of the policy with our proposed CDM.

---

> > ### Comment · Reviewer_b9og · 2025-11-26
> >
> > My concerns have not been fully addressed. Therefore, I maintain my original score.

---

> > > ### Author Response · Authors · 2025-11-26
> > >
> > > We sincerely ask the reviewer to discuss our reply one by one. In particular, which part addressed some of your concern, and which parts are not fully addressed? We are willing to continue revising and discussing the manuscript.

---

### Official Review · Reviewer_ScyS · 2025-10-29

**Soundness:** 3
**Presentation:** 4
**Contribution:** 3
**Rating:** 6
**Confidence:** 4

**Summary:**

This paper introduces Camera Depth Models (CDMs), a neural network plugin designed to improve the accuracy of depth readings from consumer-grade depth cameras (like RealSense). The authors develop a large multi-camera depth dataset (ByteCameraDepth) and a novel data engine to train the CDMs by modeling camera-specific noise in simulation. By generating accurate, "simulation-like" depth data in the real world, the CDMs effectively bridge the sim-to-real gap for geometric perception. The main contribution is demonstrating that a depth-only manipulation policy, trained purely on clean simulated depth data (without real-world fine-tuning or adding noise), can transfer seamlessly to real robots and achieve high success rates on challenging long-horizon tasks.

**Strengths:**

1. The paper is well-structured and easy to follow. The problem is clearly defined, and the figures (especially Figure 1 and 2) effectively illustrate the proposed architecture and the core contribution of bridging the sim-to-real geometry gap.
2. This work provides a generalized, plug-and-play solution that allows researchers to fully leverage clean simulation depth without the current necessity of laborious real-world noise modeling or fine-tuning, dramatically lowering the barrier for generalizable, geometry-based robot policies.

**Weaknesses:**

1. In Section 3.4, subsection "Value noise model", the notation should be $N_{\text{value}}$ and $\hat{D}\_{\text{value}}$, instead of $N_{\text{hole}}$ and $\hat{D}\_{\text{hole}}$.
2. In robotic manipulation, color information is also crucial, as in practical applications, we rarely rely solely on depth as input. I'm interested in experiments demonstrating how camera depth models benefit robot manipulation tasks when RGB information is included (i.e., using RGB-D data).

**Questions:**

1. CDM Robustness on Novel Scenes/Objects: The sim-to-real tasks test policies on randomized positions (Fig. 3), but the objects themselves seem consistent across the real and sim environments. How does the camera depth models perform when deployed on objects or scenes drastically unseen during its noise model training?

---

> ### Author Response · Authors · 2025-11-21
>
> Thank you for recognizing our work!
>
> **Regarding weakness**
> 1. Fixed
> 2. - Since our main contribution of this work is a plug-in solution for acquiring accurate depth information for real-world robotic manipulation, we decided to evaluate the strength of such improved depth inputs by ablating the RGB information.
>     - The results show that the depth-only policy with CDM is significantly better than the raw depth (as the raw depth is corrupted and uncleaned), therefore, it is reasonable that RGBD policy should also benefit from CDM compared with raw depth input (with more precise information).
>     - Training RGBD policies will further introduce the ablation test on how much the RGB part and the D part contribute; therefore, we leave it to future work to investigate how clear geometry further improves RGB policies.
>
> **Regarding questions**
>
> There is **no seen scene for CDM** during the manipulation experiments; in other words, the CDM is **not** trained on the same scene (although the policy is). It is actually an unseen scene for CDM as it is trained mainly on a set of synthetic datasets. The idea is to train the CDM to be general, to be applied to different scenes.

---

### Official Review · Reviewer_5fyz · 2025-10-31

**Soundness:** 4
**Presentation:** 3
**Contribution:** 4
**Rating:** 10
**Confidence:** 4

**Summary:**

This paper proposes to enhance the depth perception ability of different real-world depth sensor. This paper introduces the Camera Depth Models(CDMs) as a simple plugin on daily-use depth cameras, which is trained with high-quality paired data from simulation. Concretely, to generate paired depth image,  this paper first collects large-scale real-world depth-rgb images, and utilize these images to pretrain the hole noise model and value noise model to predict the low-quality depth image. With the hole noise model and value noise model, this method synthesizes camera depth for open-source simulation depth image to get paired simulation-real depth images to train the CDMs. The results prove the effectiveness of this method for depth-based manipulation policy sim-to-real adaptation.

**Strengths:**

1. This paper proposes a large-scale real-world RGB-Depth image pairs captured by seven depth cameras, which is a valuable depth-based dataset for depth prediction in the real world.

2. This paper provides a good solution to generate paired simulation-real depth images. This model enables a simulation policy to be transferred directly to a real robot for downstream tasks, achieving a zero-shot sim-to-real transfer, by simply using the CDM to clean the real camera's noise.

3. This paper provides sufficient experiments to prove that this model could achieve nearly simulation-level accuracy in depth prediction, making it possible to transfer a long-horizon simulation policy into the real-world manipulation task.

**Weaknesses:**

1. This article provides extensive visualization results and sim-to-real outcomes to validate the model's effectiveness, but it lacks some quantitative experimental results. For example, it lacks quantitative experiments on the predictive performance of the two pre-trained noise models.

**Questions:**

To what degree do these two noise models generate real-world-camera style depth images? For example, in the case of a transparent water cup, what effect does each of these models have?

---

> ### Author Response · Authors · 2025-11-21
>
> Thank you for recognizing the contribution of our work!
>
> Regarding noise models, it is tricky to directly measure the performance of the noise models, as there is no baseline to compare (even though we have some loss numbers), so our strategy in the paper is to visualize the synthesized results and evaluate the CDM trained by these noise models.
>
> To specifically answer your question, in the appendix, Fig. 12 includes real camera noises, and Fig. 13-14 in Appendix Sec. H visualize synthesized noise; by comparing them, we can observe how the hole noise model learned typical holes (like object edges), while the value noise learned the blur and typical problematic values. There is a **transparent example** from the second row of the synthesized depth images (Fig 13-14), where the problematic bottle depth is well synthesized compared to the ground truth (rightmost).

---

### Official Review · Reviewer_o2Ze · 2025-11-02

**Soundness:** 3
**Presentation:** 3
**Contribution:** 3
**Rating:** 6
**Confidence:** 4

**Summary:**

The authors propose Camera Depth Models (CDMs): Lightweight neural plug-ins for existing depth cameras that convert raw RGB + noisy depth input into denoised, metric-accurate depth — effectively making real-world perception as accurate as simulation.

Dataset contribution: it proposes ByteCameraDepth, a real-world multi-camera depth dataset comprising >170,000 RGB-depth pairs.

Compared with baseline methods, this pipeline presents better performance on benchmarks.

There are also some limitation and concerns regarding the generalization scope and model capability.

**Strengths:**

1. CDMs act as post-camera modules, requiring no changes to robot control code or simulators. Fast (< 0.15 s latency) and generalizable across camera types.
2. By producing metric-accurate depth aligned with simulation, CDMs remove the need for domain randomization and reduce reliance on real fine-tuning, unlocking large-scale sim-data reuse.
3. Large real-camera dataset + multi-sensor evaluation. Demonstrates not only numeric improvement (RMSE ↓ 40 %) but task-level transfer (0 % → 90 % success).

**Weaknesses:**

1. Generalization Scope, There are multiple aspects for generalization, for example, scene generalization, robot generalization and so on. The model is trained on some collected dataset on specific robot and scenes, can it generalize to other scenarios? Please provide more analysis about zero-shot generalization capability.

2. The previous method d3roma and FoundationStereo has different settings but also valuable to compare, can you compare the generalization capability of the proposed method and these two? especially on cases like transparent and specular area.

3. Some template issues, like Fig.5 in supp.

**Questions:**

Figure 5 in supp materials has typesetting issues.

---

> ### Author Response · Authors · 2025-11-21
>
> Thank you for your sincere feedback!
>
> **Regarding weakness**
> 1. - **Why the model is general**:
>       - a) Like the depth foundation model (PromptDA[1], DA series[2,3]), we train CDMs on a large amount of synthesized pair data to make it generalizable.
>       - b) The collected data is not limited to a specific robot/scene, but as much as possible, to train a general **noise models** that can be used to generate the training data for training CDMs (as depicted in Fig. 4) on large-scale open-source datasets. The noise model is also fine-tuned from existing strong backbones like Dino-v2[4] and DAv2[3].
>     - **Zero-shot evaluation results**
> The evaluation on the Hammer dataset is a zero-shot generalization evaluation as it is not contained in the training set.
> The qualitative visualization in Figure 8 also shows the effectiveness of zero-shot generalization of CDMs, since CDMs are trained on synthesized open-source datasets instead of the collected dataset.
> 2. - (1) **We update the quantitative comparison with D3Roma** (specifically designed for stereo IR cameras such as Realsense D435/415) on the hammer dataset, which is listed below:
>
>     | Split                | Method             | L1 ↓       | RMSE ↓     | AbsRel ↓    | $$\delta_{0.5} \uparrow$$ | $$\delta_{1} \uparrow$$ |
>     |----------------------|--------------------|-------------|-------------|-------------|---------------------------|-------------------------|
>     | **D435 (IR Stereo)** |                    |             |             |             |                           |                         |
>     |     **Ours (CDM-D435)** | **0.0258**  | **0.0404**  | **0.0312**  | **0.9842**  | **0.9951**  |
>     |     **Ours (CDM-L515)** | **0.0182**  | **0.0338**  | **0.0217**  | **0.9877**  | **0.9956**  |
>     |     D3RoMa             | 0.0338       | 0.0610       | 0.0432       | 0.9243       | 0.9740       |
>     |     Raw Depth          | 0.0550       | 0.1458       | 0.0708       | 0.9179       | 0.9543       |
>     | **L515 (D-ToF)**       |                    |             |             |             |                           |                         |
>     |     **Ours (CDM-L515)** | **0.0156** | **0.0297**  | **0.0229**  | **0.9754**  | **0.9919**  |
>     |     **Ours (CDM-D435)** | **0.0165** | **0.0349**  | **0.0246**  | **0.9613**  | **0.9855**  |
>     |     D3RoMa             | 0.0282       | 0.0522       | 0.0428       | 0.9329       | 0.9652       |
>     |     Raw Depth          | 0.0312       | 0.0813       | 0.0475       | 0.9098       | 0.9429       |
>     | **Helios (I-ToF)**     |                    |             |             |             |                           |                         |
>     |     **Ours (CDM-L515)** | **0.0248** | **0.0403**  | **0.0334**  | **0.9468**  | **0.9871**  |
>     |     **Ours (CDM-D435)** | **0.0272** | **0.0457**  | **0.0372**  | **0.9297**  | **0.9806**  |
>     |     D3RoMa             | 0.0400       | 0.0612       | 0.0523       | 0.8934       | 0.9690       |
>     |     Raw Depth          | 0.0312       | 0.0813       | 0.0475       | 0.9098       | 0.9429       |
>
>     Note that FoundationStereo is specifically designed for stereo RGB cameras (such as ZED), which cannot be measured on the Hammer dataset because the dataset does not contain stereo RGB images. We additionally compare both methods qualitatively in Appendix F. In summary, CDMs achieve much better performance than D3RoMa on the Hammer dataset and also show competitive qualitative performance compared to FoundationStereo on ByteCameraDepth dataset.
>
>    - (3) In comparison, our method is a general solution that can be used for various depth cameras, where the input of CDMs is just RGBD images.
>    - (4) Furthermore, we highlight that our plug-in solution is more lightweight and can be employed at a higher frequency compared to the diffusion solution of D3RoMa and the large network design of Foundation Stereo:
>
>         > **inference time comparison (single 4090)**
>
>         > FoundationStereo: 0.319 $\,\pm\,$ 0.002s (~3.13Hz)
>
>         > D3RoMa: 1.531 $\,\pm\,$ 0.004s (~0.65Hz)
>
>         > **CDMs: 0.151$\,\pm\,$0.002 s  (~6.67Hz)**
>
> 3.  (same as questions) Fixed.

---

### Meta-Review · Area_Chair_ED7k · 2026-01-09

**Summary:**

The paper proposes a model for improving depth maps output by RealSense cameras and uses the model to support sim-to-real transfer of robotic policies.


Based on the reviews, the rebuttal, and the paper itself, the key strengths and weaknesses of the paper are:

Pros:
1. The paper is well presented, has a reasonable story
2. Connecting the dots between perception and action is not easy and useful
3. Good performance of the proposed CDMs on depth map refinement itself
4. Good performance on manipulation policy transfer enabled by the CDMs
5. Collect a fairly large dataset of RGB-depth pairs that will be released


Cons:
1. Some related work on depth estimation/refinement is missing. On depth estimation, DepthPro is a recent state-of-the-art method that is not mentioned and not compared against. On depth refinement, I am less sure off the top of my head, but there has certainly been work on depth map refinement e.g. for Kinect fusion.
2. No baseline for sim-to-real transfer where the model is trained on synthetic data with simulated real-like noise. The authors dismissed this request in the rebuttal and I don’t think it’s fair that they did. It is a very reasonable baseline. It solves the same problem of policy transfer. The argument against including it seems to be along the lines of “we’re interested in doing what we do and not interested in doing what others do or have done”, which is not exactly how baseline selection should work. Given that policy transfer is a key aspect of the paper, it would be informative to show how clean sim -> clean real compares to noisy sim -> noisy real.

Overall, the paper is somewhat borderline: it’s good engineering and good results generally, but not a breathtaking degree of innovation, and some related work and baselines are missing. Given that the presentation is good and the whole perception-action pipeline is improved, I at this point still recommend acceptance, but very much encourage the authors to take the cons to heart and act upon them, as opposed to arguing them away.

**Reviewer Concerns:**

- Sim-noisy -> real-noisy baseline for manipulation transfer. Discussed above.
- Lack of generalization analysis for the policies -> the authors pointed out that there are some generalization experiments
- RGB-D policies -> the authors postpone this to future work, which is in principle fair. Sim-to-real transfer may be more tricky with RGB.

**Reviewer Scores:**

I would not expect the reviewers to change their ratings substantially.

---

### Decision · Program_Chairs · 2026-01-26

Accept (Poster)